# Retrosynthetic reaction pathway prediction through neural machine translation of atomic environments

Umit V. Ucak[1], Islambek Ashyrmamatov [1], Junsu Ko[2] & Juyong Lee [1,2✉]

Designing efficient synthetic routes for a target molecule remains a major challenge in organic synthesis. Atom environments are ideal, stand-alone, chemically meaningful building blocks providing a high-resolution molecular representation. Our approach mimics chemical reasoning, and predicts reactant candidates by learning the changes of atom environments associated with the chemical reaction. Through careful inspection of reactant candidates, we demonstrate atom environments as promising descriptors for studying reaction route prediction and discovery. Here, we present a new single-step retrosynthesis prediction method, viz. RetroTRAE, being free from all SMILES-based translation issues, yields a top-1 accuracy of 58.3% on the USPTO test dataset, and top-1 accuracy reaches to 61.6% with the inclusion of highly similar analogs, outperforming other state-of-the-art neural machine translation-based methods. Our methodology introduces a novel scheme for fragmental and topological descriptors to be used as natural inputs for retrosynthetic prediction tasks.

[1] Department of Chemistry, Division of Chemistry and Biochemistry, Kangwon National University, Chuncheon 24341, Republic of Korea. [2] Arontier co., Seoul, Republic of Korea. ✉email: juyong.lee@kangwon.ac.kr

Planning the reaction pathways of organic molecules is a central component of organic synthesis. The idea of reducing the complexity of a desired organic molecule by considering all logical disconnections forms the basis of the retrosynthetic approach[1–3]. Therefore, the aim of the retrosynthetic approach is to suggest a logical synthetic route to generate a target molecule from a set of available reaction building blocks. A conventional retrosynthetic approach acts recursively on a target molecule until chemically reasonable pathways are identified[4]. From a broader perspective, existing predictors for forward and backward reactions can be classified into those that rely on known reaction templates and those that are template-free, data-driven networks trained in an end-to-end fashion.

Template-based approaches use reaction templates to predict reactants from a product. Reaction templates are extracted from data using algorithms or encoded manually. For manual encoding, deep chemical expertize and management of complex transformation rules are needed[5–8]. Data-driven approaches, however, enabled automated extraction of large reaction templates from reaction data[6,9–14]. For retrosynthesis prediction, each template is applied to a product to find a match, subgraph isomorphism. If a proper isomorphism is found, a product is transformed depending on the template. This process continues until chemically reasonable pathways are identified[14].

Template-free methods have emerged as an effective means to complement the following issues of template-based methods. Exploring the space of possible reaction templates is challenging because of the vast size of chemical space. If only a limited number of reaction templates are used, template-based methods may not be able to provide novel disconnections[6,15]. On the contrary, if a large number of reaction templates are considered, the computational burden to find a proper template increases significantly. Currently, templates are either hand-crafted by experts[7] or generated from reaction databases with heuristic algorithms[9,11]. Thus, the degree of template generality/specificity can lead to either low-quality or incomplete recommendations. Lastly, reaction templates are extracted based on atom mapping, which remains a challenging issue for all template-based methods[16]. Atom mapping quality also affects model performance.

Template-free methods can be further subdivided according to the molecular representation protocol into: (i) graph-based methods[15,17–19] and (ii) sequence-based methods[16,20–22]. Sequence-based modeling recasts the problem of reaction pathway planning as a language translation problem using a string representation of molecules[23]. Most state-of-the-art forward- and backward-reaction predictors are built on the Transformer architecture[24]. Transformer is a neural machine translation (NMT) model that solely depends upon attention mechanism[24,25]. Molecular Transformer was the first adaptation of Transformer with SMILES[26] for the forward-reaction prediction task[27,28]. Further studies demonstrated the ability to make general predictions using different compound databases, including drug-like molecules[29] and carbohydrate reactions[30], to examine regioselectivity and stereoselectivity. This success has paved the way for developing retrosynthesis predictors using SMILES and Transformer[31–36].

SMILES strings are typical inputs for retrosynthetic predictors using NMT models. Despite its widespread usage, SMILES easily leads to erroneous predictions because of its fragile and complex grammar. For instance, a single character change is often enough to invalidate an entire SMILES string. Thus, SMILES-based prediction methods tend to make many grammatically invalid predictions reducing their prediction efficiency. In a recent study, the top-10 invalidity error (SMILES parsing errors) was reported as much as 12.6%[33]. To solve this problem, SCROP[34] included a neural-network-based syntax corrector to decrease the invalidity rate. Similarly, other studies[32,36] focused on determining the causes of invalid SMILES to improve the prediction accuracy. In addition, grammatically valid SMILES are not guaranteed to be semantically valid due to, i.e., explicit valence and kekulization errors. To circumvent these problems, alternative syntaxes such as DeepSMILES[37] and SELFIES[38] were developed. In our previous study[39], we demonstrated that representing molecules as the sets of fragments is an effective solution to the aforementioned problems.

Considering the complexity of retrosynthetic analysis, an efficient representation of source-target data structure is critical for accurate predictions. In this study, we show that representing molecules using sets of atom environments (AE) is an efficient alternative approach to conventional SMILES-based approaches for devising a retrosynthetic prediction model. AEs are topological fragments centered on an atom with a preset radius[40], defined by the number of shortest topological distances between atoms via covalent bonds. Unlike SMILES tokens, each AE is chemically meaningful and easily interpretable. NMT models are designed to translate between pairs of words from different languages, whereas SMILES-to-SMILES translations require a model to learn chemical changes mostly via rearrangements of SMILES tokens due to the conservation of atom types in an ideal reaction dataset. On the other hand, AEs in close vicinity of reaction center encapsulate the chemical change. The chemical change becomes observable in associated tokens, fragments, and thus can be captured by the model.

In this study, we propose a direct translation approach for a single-step retrosynthetic prediction by associating the AEs of the reactants with the products. Throughout the study, AEs are regarded as the basis of molecules and employed in our prediction workflow. Our design enables us to capture chemical changes by focusing on fragments related to the reaction centers. To accurately generate the reactant candidates for a target molecule, we used the Transformer architecture[24]. We showed that our model achieves a top-1 exact matching accuracy of 58.3%. The overall accuracy increased to 61.6% by adding extremely similar predictions. These results are better than those of the existing methods, without suffering from problems associated with the SMILES representation.

## Results

**The model framework**. Transformer connects the encoder and decoder units to translate between sequences by effectively employing a multi-head attention mechanism on each unit. Input and output sequences for our Transformer model are the lists of AEs. We tested several different schemes to convert molecules into a list of fragments, such as MACCS keys[41], the bit vectors of extended circular fingerprint (ECFP)[42], and AEs[40]. AEs are fragments consisting of a central atom and its covalently bonded neighbors with a predefined radius. They can be considered the basis of constructing molecules, in a similar manner to the pieces of a jigsaw puzzle.

An overview of our Transformer-based model, viz. Retro-TRAE, is provided in Fig. 1a. First, a product molecule is decomposed into a set of unique AEs. Each AE, described by a SMART pattern[43], is associated with a unique integer value. Lists of AEs are provided as input sequences for RetroTRAE. RetroTRAE was trained to predict the AE sequences of the true reactants.

In Fig. 1b, the string representation of benzene is given as common SMILES and SMARTS patterns representing the AEs generated by the ECFP fingerprint, along with the recently developed SELFIES[38] description. SMARTS and SELFIES are

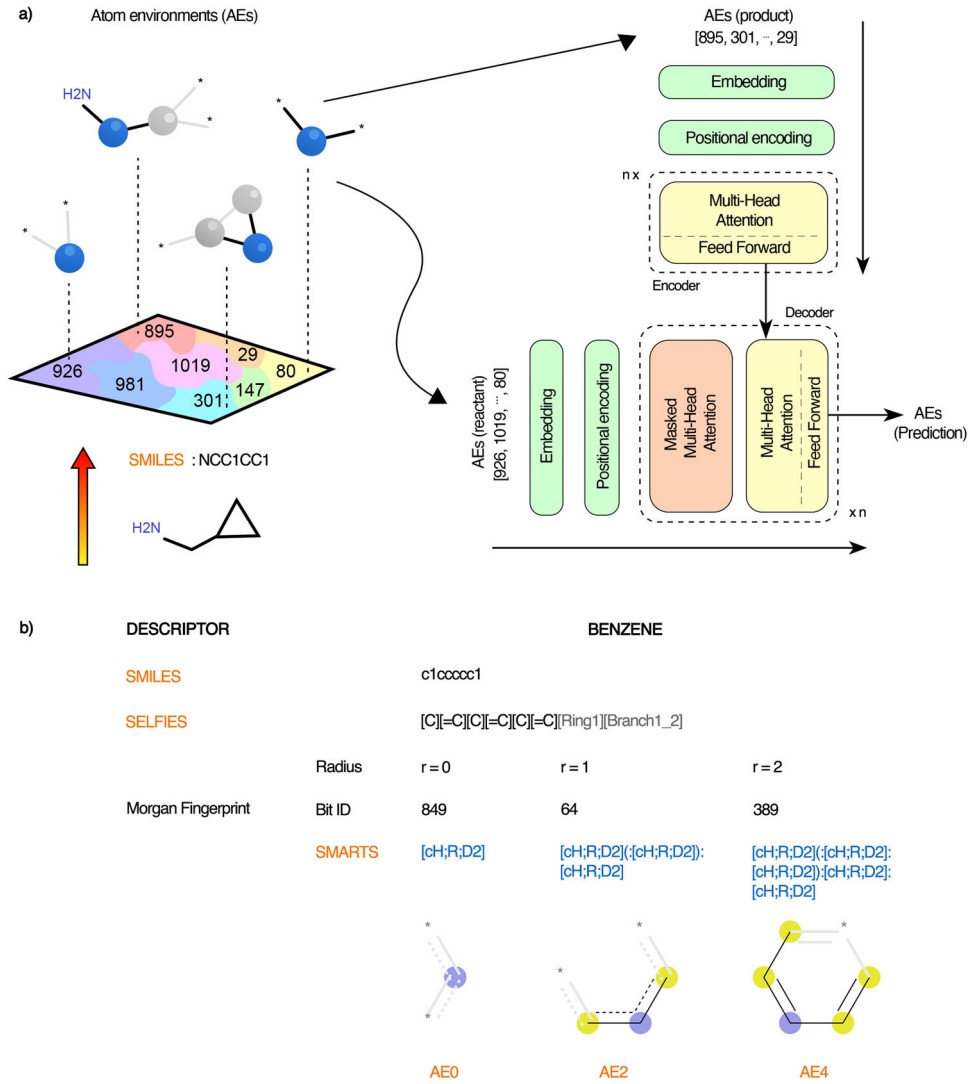

**Fig. 1 The model framework. a** A schematic of RetroTRAE including the input–output structure. **b** String representations of benzene are presented in the form of SMILES, SELFIES, and as a combination of SMARTS patterns generated by the Morgan fingerprint. In AEs renderings, the central atom is highlighted in blue whereas aromatic and aliphatic ring atoms are highlighted in yellow and gray, respectively. A wildcard [*] is used to represent any atom.

similar with respect to the level of information they display. The text sections of the SMARTS description contain two levels of detail: the first level represents the aromaticity and H count of the element, and the second level includes the number of neighboring heavy atoms and ring information (represented by "D" and "R", respectively). By definition, AEs with radius $r = 0$ only include the atoms of the central atom type. We denote the set of all AEs with $r = 0$ as AE0. AEs with $r = 1$ contain the central atom, all atoms adjacent to the central atom (nearest neighbors), and all the bonds between these atoms. The set of all AEs with $r = 1$ is denoted as AE2.

**Comparing fragmentation schemes**. We evaluated the retrosynthetic predictor performance using the selected fingerprint variants to determine the best fragment representation using the unimolecular dataset as shown in Table 1. We compared Transformer results with previously developed sequence-to-sequence fragment-based retrosynthetic predictor[39]. The Transformer-based models coupled with the ECFP representation demonstrated major improvements over previous biLSTM-based methods in terms of the exact match accuracy. This enhancement represented a substantial overall performance gain of 17–19%. The

Transformer model representing molecules with the union of AE0 and AE2 outperformed all other models, achieving an exactly matching accuracy of 55.4%. The addition of bioactively similar predictions increased the accuracy by 12.7% over the exact matches, resulting in an overall model accuracy of 68.1%.

When we used MACCS keys for fragmentation, the number of exact and bioactively similar matches were similar to that of the biLSTM-based model. This suggests that MACCS keys have low resolution power than AEs. In contrast, AE2 describes the chemical space more precisely, and provides 60 times higher resolution power than MACCS keys (Supplementary Table 2). The model using ECFP2 also performed well and showed slightly worse performance than using AEs. Hereafter, we refer to the model with the union of AE0 and AE2 as RetroTRAE.

**Optimal fragments for single-step retrosynthesis predictions**. Another interesting observation is the poor performance of ECFP4. The number of exact matches dropped to nearly a half of that of ECFP2. This poor performance may be due to a high collision rate of ECFP4 (Fig. 2). We investigated the number of unique AEs of radii 0, 1, and 2 that were associated with the activated bits of hashed ECFPs for the unimolecular reactions.

**Table 1 Performance summary of various Transformer-based models trained with different fragmentation schemes in unimolecular test set and a comparison with the BiLSTM-based models.**

|  | BiLSTM-based | | | Transformer-based | | | | |
| --- | --- | --- | --- | --- | --- | --- | --- | --- |
|  | **MACCS** | **ECFP2** | **ECFP4** | **MACCS** | **ECFP2** | **ECFP4** | **AE2** | **AE0 ∪ AE2** |
| $T_c = 1.0$ | 29.9 | 35.6 | 9.1 | 30.1 | 54.9 | 26.0 | 50.9 | 55.4 |
| $T_c \geq 0.85$ | 57.7 | 50.7 | 28.4 | 57.5 | 67.6 | 50.1 | 59.9 | 68.1 |
| $\overline{T_c}$ | 0.84 | 0.80 | 0.66 | 0.85 | 0.88 | 0.73 | 0.84 | 0.88 |

Success rates (%) are given with respect to exact and bioactively similar matches ($T_c \geq 0.85$) and the mean Tanimoto coefficients of all predictions are listed.

**Fig. 2 Optimal radius for Morgan fingerprint in a translation task.** The number of Morgan bits according to the number of unique SMARTS patterns from AE0 (blue), AE2 (cyan), and AE4 (red).

With a radii of 0 and 1, each ECFP bit contained fewer than 10 and 20 unique AEs, respectively. However, with a radius of 2, most bits corresponded to many unique AEs, ranging from 100 to 160. In other words, ECFP4 has a much higher bit collision rate than ECFP2 or ECFP0. The presence of higher-density bits complicates the relationships between the fragments of a product and the true reactants, deteriorating the prediction power of the model. Therefore, these results show that finding an optimal set of fragments representing a molecular structure most accurately is a critical factor in improving the predictive power of retrosynthesis planning.

**Performance of RetroTRAE.** Prediction performance, as a function of different similarity thresholds for RetroTRAE is given in Table 2. RetroTRAE has reached top-1 exact match accuracies of 56.4 and 60.1% trained with 10 times augmented uni- and bimolecular datasets. Augmentation slightly improved the results and stabilized the model's learning since more data and randomness were added to the network[35]. Although the AE representation is permutation invariant, the models with positional encoding perform better than those trained on without using positional information (Supplementary Table 6). This is consistent with the observation by Jaegle et al.[44].

One of the advantages of using AEs over SMILES is that a few errors do not lead to invalid predictions. Thus, we investigated how much the success rate can be improved by easing the threshold without losing functionality of the retrosynthetic framework. When single mutations (SM) were allowed, the success rates of uni-molecular and bi-molecular reactions increased to 58.1 and 60.9%, respectively. The corresponding numbers for double mutations (DM) were 60.5 and 62.7%. To quantify how low the probability of finding such extremely close

neighbors of molecules is in a large database, we performed extensive analysis by using AEs as presented in Supplementary Table 4. Considering the cumulative distribution function of AEs obtained with 1.3 million molecules in the USPTO database, only 13 pairs were found to have a $T_c$ value of 0.76 or higher. With a threshold of 0.9 or higher, most molecules in a typical database would be singletons with no near neighbors.

The mean $T_c$ of all predictions of the uni-molecular test set was found to be 0.88, which is highly statistically significant with a $p$-value $< 10^{-5}$ (Table 2). This indicates that even non-exact predictions made by RetroTRAE are still highly similar to the ground truth. Supplementary Fig. 2 shows the statistical significance of the selected similarity thresholds above which the quality of non-exact predictions is assessed in chemical terms. The inset of the figures shows the regime where $T_c$ values having a $p$-value of 0.1 (e.g., corresponds to a similarity value of 0.25 for ECFP2), whereas our lowest similarity threshold value ($T_c > 0.8$) had a $p$-value of 1e–04 or lower. Therefore, the predictions satisfying $T_c > 0.8$ occur in the high similarity regime.

**Investigation of AEs-similarity relationship.** AE formalism offers a higher resolution power than other fingerprints. This feature is particularly useful in terms of the context of fingerprint dependency of soft thresholds, Tanimoto coefficient. To demonstrate, we generated the similarity value distributions of various structural fingerprints available in RDKit using 1.3 million molecules in the USPTO dataset (Supplementary Fig. 3). For instance, within a region where a $p$-value is greater than 0.01 (equivalent to $T_c \leq 0.32$ with unified AEs), Avalon, MACCS keys, RDKit and Atom pairs fingerprints all yielded higher $T_c$ values. Topological torsion was the only exception and yielded slightly lower similarity values than AEs. These results indicate that chosen cutoffs based on AEs lie at a lower similarity level and statistically more significant than other fingerprints.

To quantify the resolution power of AE in high similarity region, two of the commonly used substructural fingerprints, MACCS and RDKit fingerprint, were compared against AEs (Supplementary Table 5). We randomly selected 10 singly and 10 doubly mutated predictions and compared the mean pair-wise similarities with respect to ground truth and the number of equivalent representations. The mean $T_c$ for AEs was 0.91, while almost none of the mutations were detected by MACCS keys. Seventeen out of 20 pairs were structurally equivalent. The RDKit fingerprint yielded a mean pair-wise similarity of 0.97. These show that the predictions obtained by hard thresholds, SM and DM, are at an exceptional level.

**Model interpretability.** It is often difficult to attribute meaning to the outcomes of deep learning methodologies. We investigated attention weights to uncover what our model actually learns. We identified that our model successfully learned the changes in chemical environments around reaction centers. In contrast to our work, in SMILES-to-SMILES translations chemical changes mostly occur via rearrangements of SMILES tokens rather than

**Table 2 The prediction accuracy (%) of RetroTRAE using x10 augmented uni- and bi- molecular reactions.**

| Datasets | $T_c = 1.0$ | SM | DM | $T_c \geq 0.85$ | $T_c \geq 0.80$ | $\overline{T_c}$ | $\overline{S}$ |
|---|---|---|---|---|---|---|---|
| Unimolecular | 56.4 | 58.1 | 60.5 | 68.2 | 72.5 | 0.88 | 0.94 |
| Bimolecular | 60.1 | 60.9 | 62.7 | 64.3 | 66.7 | 0.79 | 0.88 |
| RetroTRAE (Total) | 58.3 | 59.5 | 61.6 | 66.3 | 69.6 | 0.84 | 0.91 |

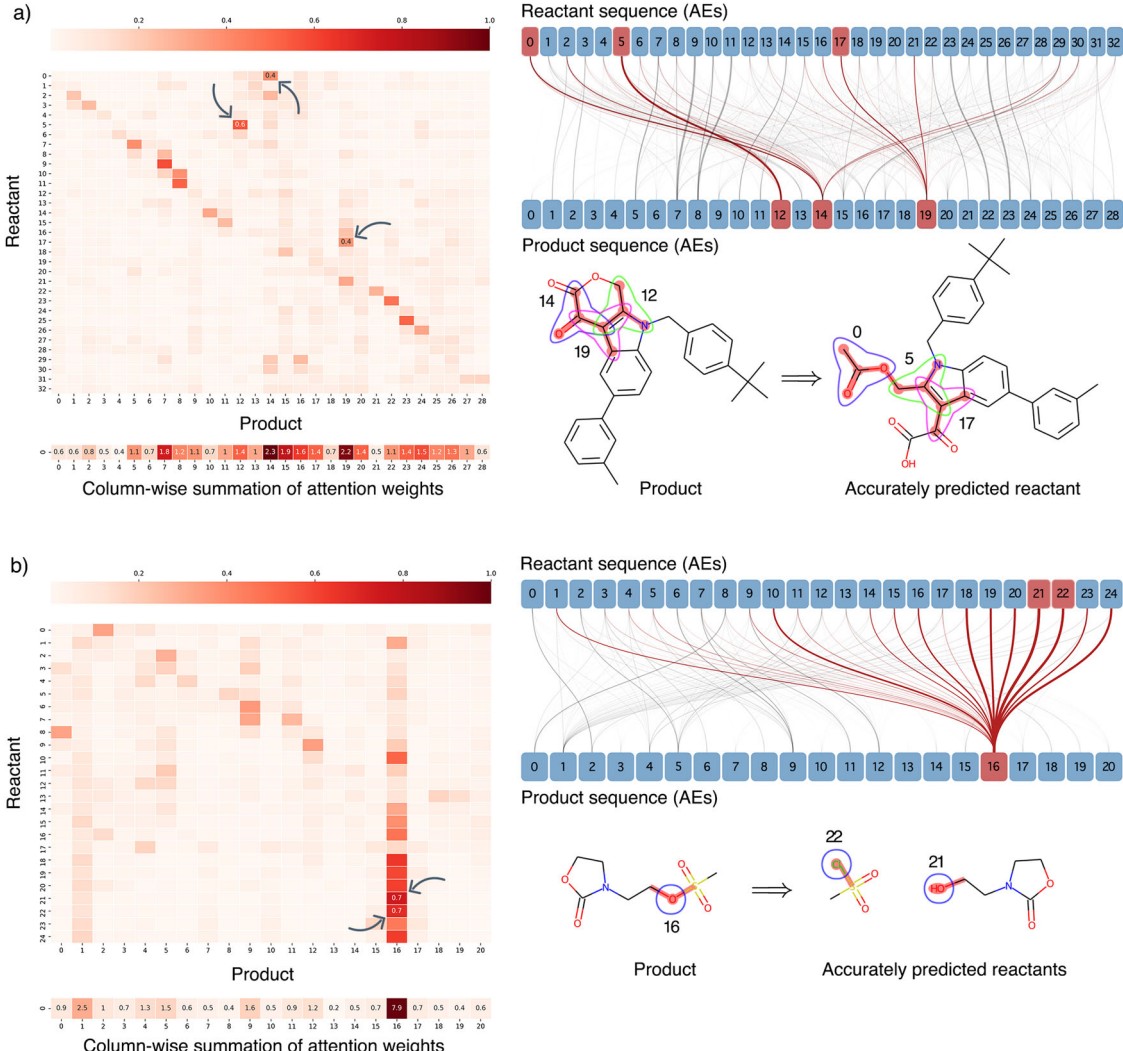

**Fig. 3 Visualization of decoder attention and interpretability of RetroTRAE.** Attention weight matrices, column-wise attention sums, and attention mappings are displayed for **a** Uni-molecular ring-opening reaction **b** Bi-molecular dissociation reaction. The AE pairs with highest attention values correspond to the reaction centers and disconnection sites. Highly correlated AE pairs between reactant and product sequences are visualized in attention maps. The widths of connections are proportional to attention values and the altered AEs surrounding the reaction center with high attention scores are highlighted.

actual transformations of chemically meaningful tokens, which hampers chemical interpretability and explainability. To address this issue, Kovács et al. proposed a framework to interpret the results of Molecular Transformer[45].

The attention weight matrices and the fragments with the highest attention values of two example reactions are visualized in Fig. 3. The AE that undergoes a change during the reaction has the highest attention value with its changed counterpart. Likewise, the AEs that remain intact tend to have highest attention with itself. The column-wise summations of attention weights

indicate the mostly attended AEs of a product by RetroTRAE. To show this, we highlighted the AEs in products that changed during the reactions and their attentions in the reactant side. Indeed, the model pays more attention to altered AEs near the reaction centers as exemplified with ring opening and dissociation reactions. These examples clearly show that AE tokens are chemically meaningful and fully interpretable by themselves as opposed to SMILES tokens.

RetroTRAE operates at the level of AEs predicting transformations from products to reactants in a single-step similar to the

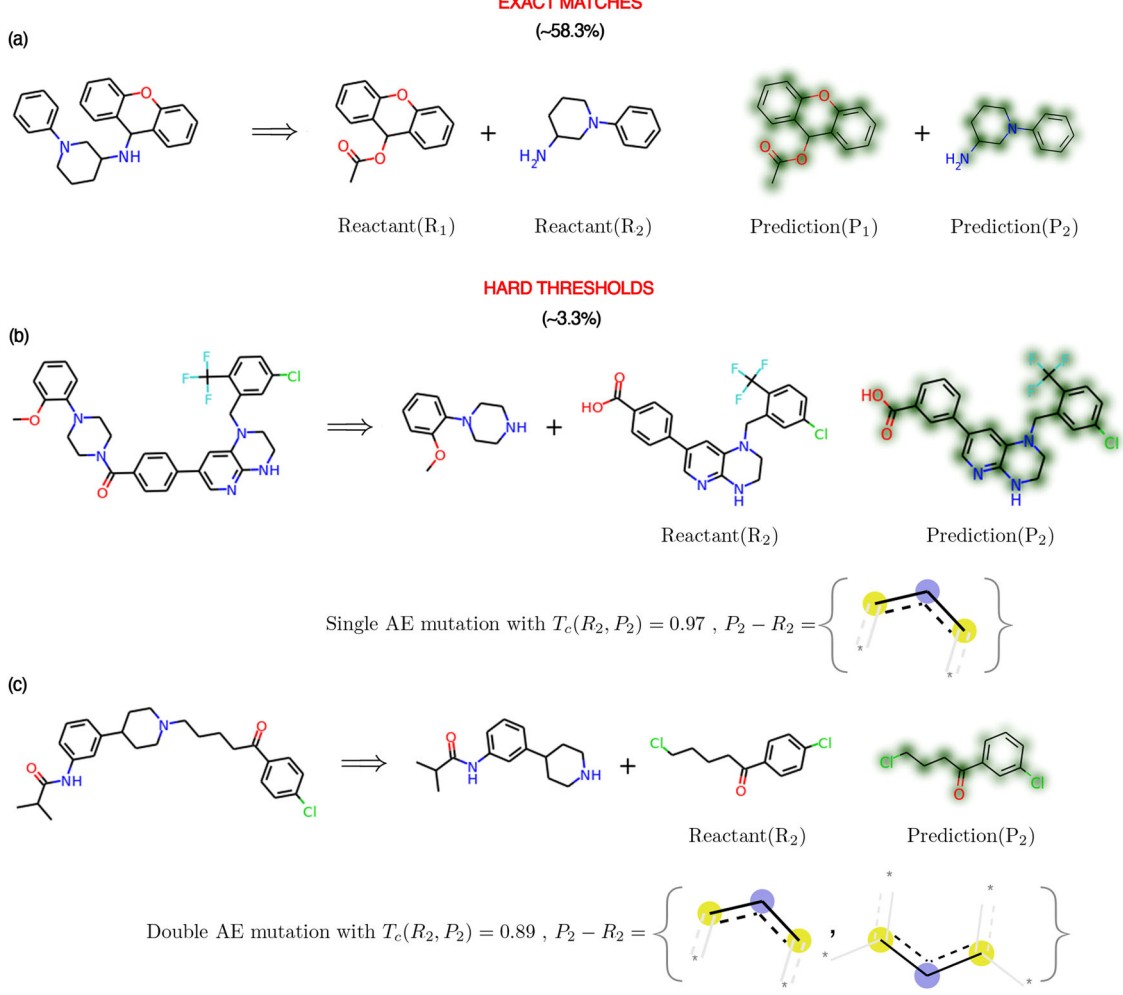

**Fig. 4 Example of RetroTRAE predictions.** Representative examples of **a** exact predictions, **b** predictions with a single and **c** double fragment mutations are shown. RetroTRAE predicted 58.3% of test set exactly. Considering highly similar predictions with single and double mutations increased the success rate by 3.3%. Distinct fragments are given as SMARTS patterns. Predictions are drawn as similarity maps using the Morgan fingerprints. For hard thresholds, the first reactant is predicted correctly and the qualities of the second reactants are evaluated. The fragments only belonging to the prediction or its true counterpart are given as set notation differences, which allows us to describe the chemical change more concretely. Colors indicate atom-level contributions to the overall similarity (green: increases in similarity score, red: decreases in similarity score, uncolored: has no effect).

previous studies[28,33,35]. The main reason for focusing single-step reactions is that the mechanistic descriptions of reactions are not provided in the USPTO database. However, there is no intrinsic limitation for the model to predict multi-step synthetic routes. The model would be able to predict multi-step synthetic routes, when it is combined with a proper search algorithm, such as Monte-Carlo tree search[13,14]. In its current form, RetroTRAE can be used in any single-step of a multi-step retrosynthesis[13].

**Examples of retrosynthesis predictions**. In addition to exact predictions, we investigated how much singly and doubly mutated predictions are similar to the ground truth. The first example illustrates an exact prediction (shown in Fig. 4a). RetroTRAE predicted 58.1% of the reactions in the test set accurately. The single and double fragment mutations together account for 3.3% of the total predictions. In single mutation cases, atom and connectivity types must be preserved, therefore only two types of structural changes are possible. First, a new environment may appear (or an existing environment may disappear) due to a misplaced single environment (e.g., at the ortho/para/meta position). With this change, all connected atom types must be preserved (Fig. 4b). Second, a single existing AE can be added or subtracted at terminal sites. Double mutations are characterized by a misplaced branching AE or a single atom substitution (Fig. 4c). If a mutation happens in the middle of a molecule, the AE centered at the mutated site and its direct neighbors are highly likely to be changed, leading to at least three AE mutations.

As indicated in the similarity maps of hard thresholds, none of the atoms of the reactant candidates negatively contributed (red) to the similarity value. With the AE representation, the length of simple aliphatic chains might be incorrectly predicted, because the length of an aliphatic chain cannot be accurately described using a set of unique fragments. Based on this observation, SM and DM predictions are much more similar to a ground truth than conventional structural analogs implying differences in certain substructures, functional groups, or several atom types. We believe that these small discrepancies are easily amendable through a visual comparison with a product. When soft thresholds are used, several AEs can be altered, making the generalization of errors highly difficult. After inspecting the bioactively similar predictions (see Supplementary Fig. 4), we concluded that the most significant aspects of retrosynthetic analysis, such as bond disconnections, reactive functional groups, and core structures, were correctly

**Table 3 A comparison of reported top-1 accuracies of retrosynthesis prediction models without additional reaction classes.**

| Model | Top-1(%) |
|---|---|
| *Non-transformer* | |
| Coley et al., similarity-based, 2017[72] | 32.8 |
| Segler et al.,–rep. by Lin, Neuralsym[a], 2020[6,33,73] | 47.8 |
| Dai et al., Graph Logic Network[a], 2019[73] | 39.3 |
| Liu et al.,–rep. by Lin, LSTM-based, 2020[16,33] | 46.9 |
| Genheden et al., AiZynthfinder, ANN + MCTS[a], 2020[14,49] | 43–72 |
| *Transformer-based* | |
| Zheng et al., SCROP, 2020[34] | 41.5 |
| Wang et al., RetroPrime, 2021[48] | 44.1 |
| Tetko et al., Augmented Transformer, 2020[35] | 46.2 |
| Lin et al., AutoSynRoute, Transformer + MCTS, 2020[33] | 54.1 |
| RetroTRAE | 58.3 |
| RetroTRAE (with SM and DM) | 61.6 |

The results are based on either filtered MIT-full[46,47] or MIT-fully atom mapped[15] reaction datasets.
[a]Reaction templates were used.

predicted. Nevertheless, we were unable to generalize the characteristics of the predictions beyond DMs, albeit within the bounds of bioactive similarity space.

**Comparison with existing retrosynthesis planning methods.** Table 3 presents a performance comparison of RetroTRAE with the existing retrosynthesis models trained without reaction class information. For a fair comparison, we compared RetroTRAE with the models that were trained and tested with the USPTO-based datasets[15,46,47]. Our approach achieved an average top-1 exact matching accuracy of 58.3%, outperforming existing NMT-based template-free models. The inclusion of single and double fragment mutations, corresponding to 3.3% of the predictions, increased the overall performance of our model to 61.6%, exceeding all current state-of-the-art performance levels. This clearly demonstrates that AEs are useful and informative representation of a molecule.

Performance differences in the SMILES-based Transformer models are attributed to improvements in data augmentation (with non-canonical SMILES)[35,48], tokenization scheme (character or atom level)[31,33], and postprocessing (by rectifying invalid SMILES)[32,34]. The better prediction accuracy of our model appears to be due to better reaction representation beyond the standard SMILES. For a comparison with top performing template-based models, we listed the top-1 accuracy of AiZynthfinder[14]. The accuracy was reported as a range of 43–72% on the filtered USPTO dataset depending on the sizes of template libraries that were used to train template prioritization models[49]. Segler and Waller reported a top-1 accuracy of 50.1% using Reaxys[13]. It should be noted that each template-based model used different training/test datasets and template extraction methods, which affect model's performance.

**Covering chemical space with atom environments.** Because AEs can be considered the basis of molecules, we investigated the number of AEs are required to represent chemical space properly. We generated the AE0 and AE2 sets using all compounds in PubChem (111 M), ChEMBL (2.08 M), and the USPTO 500 K (1.3 M) dataset and visualized their diversity and coverage (Fig. 5). Coverage was defined as the chemical space spanned by these unique AEs. The area-proportional Euler graph demonstrates that the AEs of the reactants in the USPTO dataset is not enough to describe diverse

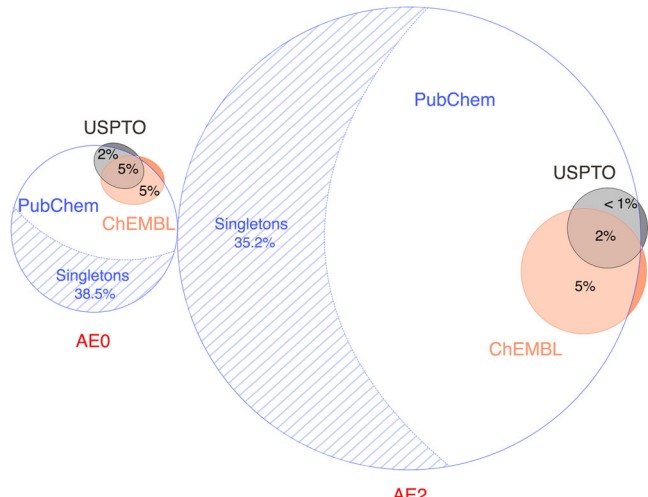

**Fig. 5 Area-proportional Euler graph representing the space of atomic environments.** The following databases are used: PubChem 110 M, ChEMBL 2.08 M (ChEMBL v28, as of May 2021), and USPTO fully atom-mapped 500 K reactions (~1.3 M molecules). AE0 is upscaled by 20 times for better visual interpretation.

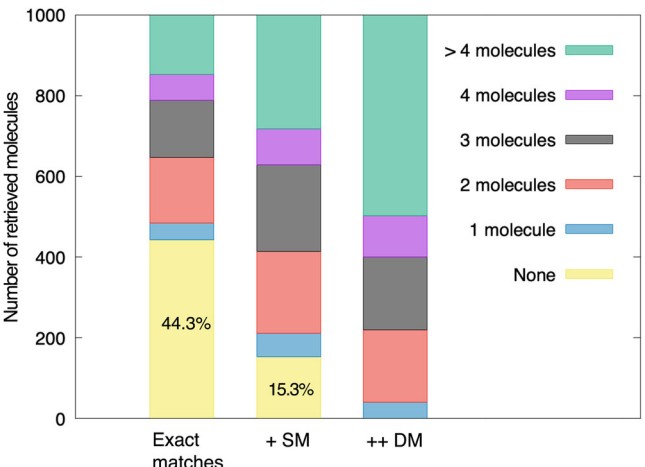

**Fig. 6 Retrieval of reactant candidates via a large PubChem compound search database.** SM and DM represent single mutation and double mutations.

molecules and do not span a broad range of chemical space. This indicates that the current USPTO reaction dataset is not large enough to train a truly general retrosynthesis predictors. We believe that our model would perform more accurately, if we have more diverse reaction datasets.

The USPTO reaction dataset contains 275 and 15,982 unique AE0 and AE2 tokens, respectively. ChEMBL and PubChem contain unique 386 AE0, 39,149 AE2, and 3450 AE0, 533,276 tokens, respectively. Although there are large differences in favor of PubChem, a significant portion of these unique AEs occurs only once in the whole set, which we refer to as singletons. The percentages of singletons were 38.5 and 35.2% for the AE0 and AE2 sets generated from PubChem. The cardinality of each set of unique AEs was supplied as Supplementary Note 1 together with their intersections.

**Retrieving reactant candidates via atom environments.** After predictions are made by RetroTRAE, the chemical structures of the predicted reactants, the set of AEs, can be retrieved through a

database search. We investigated the success rate of retrieving a reactant candidate with 1000 USPTO test molecules using Pub-Chem. The retrieval test results showed that more than half the predictions (55.7%) could be retrieved accurately (Fig. 6). Allowing SM increased the retrieval rate by ~30%. When DM were allowed, all the test molecules could be retrieved success-fully. In other words, the predictions of RetroTRAE can be restored to real molecules exactly or highly similar molecules with a discrepancy of two AEs at most. As mentioned previously, molecules with SM and DM generally have differences in ste-reochemistry, the length of their aliphatic chains, and the location of their peripheral functional groups, such as ortho/meta/para positions (Fig. 4). These results suggest that representing and predicting molecules with AEs is a viable and practical approach.

Finally, it is worth mentioning that AEs are less degenerate, i.e., have fewer reactant candidates corresponding to a prediction, than ECFP fingerprints during the retrieval process. Using ECFP bit indices for database searches retrieve 1.7 times more reactant candidates on average. The difference is mainly due to bit collisions that occur during truncation to the bit vector and the absence of stereochemical information in our dataset.

## Discussion

We developed a new template-free retrosynthesis prediction model, namely RetroTRAE, using the Transformer architecture and the AE representation. RetroTRAE provides fast and reliable retrosynthetic route planning for substances whose fragmentation patterns are revealed. We demonstrated that AEs are promising descriptors for developing other generative and sequence-based architectures in addition to conventional SMILES-based approa-ches. Using AEs has advantages compared with conventional SMILES-based models. First, it need not learn complex grammar of SMILES. Second, each token is an actual substructure of a molecule making a model more interpretable in a chemical sense. Third, no atom mapping procedure is necessary, which can be computationally expensive and introduce additional errors to input data. Detailed analysis of predictions including attention values suggests that models trained with AEs are fully inter-pretable and AEs with high attention values reveal reaction centers.

RetroTRAE showed comparable or improved performance compared to other state-of-the-art models. We critically assessed the retrieval process that converts a set of fragments into a molecule with respect to coverage, degeneracy, and resolution. RetroTRAE predicted reactant candidates with an exact match accuracy of 58.3%. In addition to the exact match accuracy, highly similar reactant candidates with single and double mutations were exceptionally similar to ground truth with a $p$-value $< 10^{-7}$. The overall accuracy with singly and doubly mutated predictions was 61.6%, outperforming current state-of-the-art methods. We emphasize that this comprehensive study addresses the major limitation of structural fingerprints, which precludes their implementations in NLP models. We believe that our findings will open new possibilities for the development of NMT models for chemistry using sequential data, not only for retrosynthetic prediction but also for reaction and property predictions.

## Methods

**Atom environments**. We employed the concept of circular AEs to represent the molecules in the reaction dataset. Circular environments are defined as topological neighborhood fragments of varying radii containing all bonds between the included atoms[40]. They are centered on a particular atom, called the central atom. The radius refers to the maximum allowed topological distance between the central atom and all covalently bonded atoms. The topological distance between two atoms was measured as the number of bonds on the shortest path between them. Thus, an AE of radius r contains all the atoms in a molecule with a topological distance r or smaller from the central atom, and all bonds between them.

To construct the AEs, we used the ECFPs of varying radii implemented in RDKit. We extracted all unique fragments that were folded into the bits of ECFPs. AEs generated by the ECFP algorithm are invariant to rotation and translation and are easily interpretable as SMARTS patterns[50,51]. The AE representation does not record any connectivity information. Thus, there is no one-to-one correspondence between molecular structure and the set of AEs. In our analysis, we considered AEs as the pieces of a molecular jigsaw puzzle. Larger pieces (higher radii fragments) encompass small pieces (smaller radii fragments). A proper fingerprint radius ensures that a fragment isomorphic to the molecular structure can be found (Supplementary Fig. 1). However, as discussed in Section "Optimal fragments for single-step retrosynthesis predictions", the optimum AE radius for a neural translation task is equal to 1.

We focused on two fragmentation schemes: AEs and ECFPs. A word-based tokenization scheme was applied to both AEs and the indices of the ECFP bit vectors. An ECFP bit vector corresponds to a one-hot encoded vector in fingerprint space, such as a sentence, which is one-hot encoded in vocabulary space. In this study, the following representations encoded as bit indices and SMARTS were tested:

- AE0 and AE2 corresponding to AEs of radius 0 and 1,
- ECFP0, ECFP2, and ECFP4[42] corresponding to the Morgan fingerprints of radius 0, 1, and 2, hashed into a dimension of 1024.

AEs of radius 2 (AE4) result in millions of distinct fragments. Because of the vast vocabulary size of AE4, they are not suitable for translation purposes. Thus, only the hashed version of the Morgan fingerprint was selected for a radius of 2.

**Dataset**. To evaluate and compare our model with the current state-of-the-arts, we used the subset of the filtered US patent reaction dataset, USPTO-Full, obtained with a text-mining approach[46,47]. This subset[15] contains 480 K atom-mapped reactions after removing duplicates and erroneous reactions from USPTO-Full. Preprocessing steps to remove reagents from reactants are described in refs. [16,22], which were based on a ">" token in reaction SMILES. By following this procedure, Zheng et al. provided canonicalized reactant and product SMILES[34]. In addition, there was no reaction class information available in this dataset.

We used Zheng's version of USPTO and carefully curated the product-reactant pairs. We limited ourselves to single product reactions, corresponding to 97% (465 K) of all the available reactions. We then omitted multi-component reactions primarily because they occupy less than 1.65% of the whole dataset. We set an upper length limit for sequences up to 100 fragments. In this study, we have not used any atom-to-atom mapping algorithm. With forward reactions, we ended up two distinct curated datasets based on the number of reactants, consisting of unimolecular (R $\longrightarrow$ P) and bimolecular (R$_1$ + R$_2$ $\longrightarrow$ P) type reactions, with a combined size of 414 K. Since retrosynthesis implies an abstract backward direction, we named our datasets unimolecular and bimolecular reactions. Additionally, we used the PubChem compound database including 111 million molecules, and the ChEMBL database to recover molecules from a list of AEs and compare the space of AEs[52,53].

**Training details**. Our curated datasets were randomly split into a 9:1 ratio to generate the training and testing sets. The validation set was randomly sampled from the training set (10%) prior to training and used only for optimizing the hyperparameters. We used the Adam optimizer[54] to train model parameters in combination with a negative log-likelihood (NLL) loss function. The best hyper-parameters were chosen according to their performance on the validating set. With these hyperparameters, the average training speed was approximately 12 min per epoch with a batch size of 300 on a single Quadro RTX 8000 card. We applied dropout with a rate of 0.1[55].

The open-source RDKit[50] module version 2020.03.1 was utilized to generate ECFPs and AEs. The PyTorch[56] machine learning library was used for constructing and training the model. The model was configured similarly to the original Transformer paper, except the normalization layer was applied prior to self-attention, multi-head attention and feed-forward operations, respectively. The outputs of the encoder and decoder were also normalized. Word-wise tokenization was applied by using the SentencePiece tokenizer[57]. The details of our key hyperparameters and hyperparameter space are described in Supplementary Table 1.

**Evaluation procedure**. To evaluate the performance of our translation model, a suitable metric was required to measure the similarity between predictions and the true reactants. The Tanimoto ($T_c$) and the Sørrensen–Dice coefficient ($S$) as two of the special cases of the Tversky index were the similarity metrics used in this study. The exact form of the Tversky index is as follows:

$$S(X, Y) = \frac{|X \cap Y|}{|X \cap Y| + \alpha |X - Y| + \beta |Y - X|} \qquad (1)$$

Here, $\alpha$, $\beta \geq 0$ are the parameters of the Tversky index. Setting $\alpha = \beta = 1$ leads to the Tanimoto coefficient; setting $\alpha = \beta = 0.5$ leads to the Sørrensen–Dice coeffi-cient. The Tanimoto and Dice coefficients measured between two molecules range between 0 and 1. The value of zero represents the total dissimilarity, whereas a

value of 1 represents the exact match. We used the ccbmlib Python package[58] to generate the similarity value distributions of the fingerprints and assess the statistical significance of the Tanimoto coefficients. This implementation also allowed for a quantitative comparison of similarity values between various fingerprint designs.

Unlike SMILES-based methods, small prediction errors of the AE representation do not yield invalid predictions. Thus, multiple degrees of accuracy can be calculated due to the native design of our model. The results were computed with four different cutoffs, which can be categorized as: (a) hard thresholds, and (b) soft thresholds. We define hard thresholds as the discrepancies of one or two fragments. We call arbitrary thresholds based on the Tanimoto coefficient soft thresholds such as $T_c \geq 0.85$. These measures are conventionally used to screen similar molecules. For example, molecules having $T_c \geq 0.85$ tend to exhibit similar biological activities[59–67]. This assumption has been tested in multiple studies with different datasets and fingerprints[65,67–70].

Hard thresholds offer the following advantages over soft thresholds. First, hard thresholds do not depend on sequence length (Supplementary Table 3). Second, contrary to soft thresholds, they allowed us to easily find the type and number of fragments that deviated from the ground truth. Finally, by using hard thresholds, we can avoid any risk of losing high-quality reactant candidates that could be excluded with soft thresholds. The structural complexity of a molecule is closely associated with a fingerprint length. This suggests that high-quality predictions with low and medium complexity, relatively smaller molecules, have a higher chance of being excluded by soft thresholds. For example, a high-quality double mutated prediction with medium complexity represented with 13 AEs could be overlooked by a bioactively similar threshold ($T_c \geq 0.85$).

In this study, we used top-1 predictions as the best recommendations to report the performance of model, as well as for molecular search and retrieval. Since there are many ways to decompose a molecule, retrosynthetic prediction tools can procure many different possible synthetic routes. However, the analyses showed that only 6% of the USPTO dataset has at least two sets of reactants[46,47,49]. Thus, using top-1 accuracy is a legitimate measure to assess a single-step retrosynthesis predictor trained on the USPTO dataset. Top-N accuracy for evaluating retrosynthesis prediction has recently been disputed because, with each prediction, a model tends to find the next frequently observed answer among reactions in a dataset rather than making a chemically more meaningful prediction[28,49]. A few alternative metrics were newly suggested, such as Round-trip[28], and MaxFrag[35].

## Data availability

The data that support the findings of this study are generated by using Zheng's version of USPTO dataset and are available in the RetroTRAE GitHub repo: https://github.com/knu-lcbc/RetroTRAE. Source data are provided with this paper.

## Code availability

The source code of this work and associated trained models are available at the RetroTRAE GitHub repo: https://github.com/knu-lcbc/RetroTRAE[71].

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

## Acknowledgements

This work was supported by Arontier co. This work also was supported by the National Research Foundation of Korea (NRF) grants funded by the Korean government (MSIT) (Nos. NRF-2019M3E5D4066898, NRF-2018R1C1B600543513, and NRF-2020M3A9G7103933 to I.A. and J.L.). This work was supported by the Korea Environment Industry & Technology Institute (KEITI) through the Technology Development Project for Safety Management of Household Chemical Products, funded by the Korea Ministry of Environment (MOE) (KEITI:2020002960002 and NTIS:1485017120 to U.V.U. and J.L.).

## Author contributions

U.V.U. and J.L. conceived and designed the study. U.V.U. and I.A. processed data, trained the models, and analyzed results. U.V.U., I.A., J.K., and J.L. discussed and interpreted the results. U.V.U., I.A., and J.L. wrote the manuscript.

## Competing interests

The authors declare no competing interests.
