## [Peer Review File · Nature Communications]

REVIEWER COMMENTS

Reviewer #1 (Remarks to the Author):

The paper by Ucak et al. report a transformer model for single step retrosynthesis prediction. While previous such transformer models learned to translate the SMILES line notation of the reaction product to the SMILES notation of the reactants, the present model learns to translate atom environments (AEs) extracted from the ECFP fingerprint of the reaction product to those of the reactants, and then reconstructs the reactants from the predicted AEs. AEs are interpreted as SMARTS for illustration in figure 1.

The whole approach seems a bit fuzzy as it is well known that the correspondance between ECFP AEs and molecular structure is not unequivocal. Indeed, the model does not perform particularly well, and the author define a "bioactively similar" concept to increase success by stating that if a predicted reactant is similar to the correct one that's OK. This approach unfortunately reveals a profound misunderstanding of what retrosynthesis is about: one cannot synthesize products from approximate analogs, and the examples in Figure 4 illustrate the problem: there are mistakes in the position and number of aromatic halogens in the predicted reactants.

In summary, the presented method is flawed and the performance reported is not convincing. The work does not represent a significant advance in the field, is unsuitable for a general interest journal, and should be revised in depth and submitted to a specialized cheminformatics journal.

Detailed comments:

1. The authors should be clear in that they are only limiting themselves to the task of single step predictions and not predicting full retrosynthetic pathways. This differentiation is useful for the reader to know the context and applicability of the work carried out.
2. Abstract: What are the issues of SMILES-based retrosynthesis? This is not clear for a non-expert in the field.
3. Introduction: The authors state that the template-free methods have emerged as means of addressing methodological issues with template-based retrosynthesis. An outline of where the

authors think improvements are made should be given, as there are specific regions in which template-based methods overcome methodological challenges, and this is not a general problem.

4. Line 24, page 3 – The word synthetically accessible should be removed as SMILES is not synthetically accessible and usage of the term can lead to confusion.

5. End of page 3 – It has been shown that the transformer learns pairwise relationships between atoms in the reactants and products. In light of this what additional benefit does tokenisation of atomic environments add?

6. Line 12 page 4 – The problems with SMILES could be clearly outlined in the introduction to aid the readers in understanding why fragment based approaches may have an advantage.

7. Methods: The authors do not state how they identify uni- and bi- molecular reactions. The reaction datasets contain a multitude of species on the reactant side of the equation, and not all species are reactive or contribute atoms to the product. How do they identify which ones are relevant from the reaction? Presumably this requires a curation step for which the details are not given.

8. Methods: the approach is limited to uni- and bi- molecular data. Can it be extended further to multi component reactions that are prevalent in reaction datasets? If so why have the authors omitted this from the study?

9. Methods: did the authors separate the validation set prior to training or did they sample the validation set from the training set after training? It is not clear if the validation set was trained on leading to a potential overfitting.

10. Methods: What libraries/packages were used to train the transformer models? What parameter values and configuration settings were used? What tokenisation scheme was used?

11. The authors state that non-transformer models do not perform as well as transformer based models for the task of retrosynthesis. Is there any evidence to support this beyond the top1 accuracy that is not a suitable metric for assessing the task of retrosynthesis? As there are many possible

disconnections that lead to the product (i.e. many reactions that can be used) top 1 accuracy fails to account for the other viable disconnections.

12. It's not possible to discuss retrosynthesis without citing the Waller 2018 Nature paper, the AiZynthfinder of Genheden et al., and the Chematika/Synthia program and many applications by Grzybowski and coworkers (even if not AI based). The influence of datasets on retrosynthesis planning performance (Thakkar et al.) must also be cited and taken into account.

13. The claim that transformers are better than template-based is entirely unfounded and not supported by the analysis in Table 3. In Table 3, in the comparison of transformer-based retrosynthesis, please a comparison with the model of ref. 15 Schwaller et al. and with the AiZynthfinder must be included.

14. The format of all references is completely wrong.

Reviewer #2 (Remarks to the Author):

The article deals with important research directions as retrosynthesis prediction; it is well written and structured. There is also a live GitHub repository supporting the work. However, I have two questions for the authors that should be answered before the final publication.

1. If a molecule tokenized with Atomic Environments instead of SMILES letters is fed to an NLP model, there must be an order and a rule to construct this order from single AEs. There are places in the article where the authors speak about sets, but the final model also has the positional encoding layer; therefore, the position is valuable. The model should have the same problems that other scientists are trying to prevent with canonical/augmented SMILES. The authors should undoubtedly clarify this issue.

2. Biosimilarity brings only higher values in the tables and nothing more. It is known that 0.85 works for biosimilarity. However, chemists face many more simple reactants (they might be structurally

similar, but the similarity coefficient maybe even be < 0.60). In a general model for retrosynthesis, such biased criteria should be avoided.

Reviewer #3 (Remarks to the Author):

The authors are proposing to use atomic environments for transformer based synthesis prediction.

This is an interesting idea that I believe has been underexploited.

However, I believe the current version needs to be significantly improved to be publishable.

1. Not only the datasets but the code needs to be open sourced so the results can be reproducible.

2. Most references are wrongly formatted

3. Important work by Alpha Lee et al related to the quality of the USPTO and interpretability of chemical reaction is ignored see for instance published earlier this year in Nature Communications <https://www.nature.com/articles/s41467-021-21895-w>

4. The Bioactivity similarity idea doesn't make any sense. The cut-off chosen, here set to 0.85 is highly dependent on which finger print is used. And in practice one wants to synthesize a specific compound not a similar compound that might already have been synthesized or is not interesting due to various reasons.

Reviewer 1

The paper by Ucak et al. report a transformer model for single step retrosynthesis prediction. While previous such transformer models learned to translate the SMILES line notation of the reaction product to the SMILES notation of the reactants, the present model learns to translate atom environments (AEs) extracted from the ECFP fingerprint of the reaction product to those of the reactants, and then reconstructs the reactants from the predicted AEs. AEs are interpreted as SMARTS for illustration in figure 1. The whole approach seems a bit fuzzy as it is well known that the correspondence between ECFP AEs and molecular structure is not unequivocal. Indeed, the model does not perform particularly well, and the author define a “bioactively similar” concept to increase success by stating that if a predicted reactant is similar to the correct one that’s OK. This approach unfortunately reveals a profound misunderstanding of what retrosynthesis is about: one cannot synthesize products from approximate analogs, and the examples in Figure 4 illustrate the problem: there are mistakes in the position and number of aromatic halogens in the predicted reactants. In summary, the presented method is flawed and the performance reported is not convincing. The work does not represent a significant advance in the field, is unsuitable for a general interest journal, and should be revised in depth and submitted to a specialized cheminformatics journal.

Reply: We appreciate the reviewer’s constructive suggestions that have helped us improve the manuscript’s clarity. We thoroughly revised it and tried to emphasize the contribution of our work to related fields. Before providing a stepwise reply to the reviewer’s comments, below we respond to specific points mentioned in the overall assessment.

1. *"The whole approach seems a bit fuzzy as it is well known that the correspondence between ECFP AEs and molecular structure is not unequivocal."*

We agree with reviewer’s comment that there is no one-to-one correspondence between molecular structure and the collection of AEs extracted from the molecule. AE representation does not record any connectivity information like other fingerprints. Here, we emphasize that using the hard threshold criteria, singly and doubly mutated predictions, mitigates the specified ambiguity.

As seen in our retrieval results in Figure 6, the probability of finding extremely close neighbors of molecules (up to two AE differences) is extremely low in the PubChem DB containing 111 million molecules. To be exact, we performed an analysis by using AEs as presented in *Appendix Table 1*. (Supplementary Table 4) Considering the cumulative distribution function (CDF) of the similarities of AE pairs obtained by using 1.3 million molecules in the USPTO database, the number of pairs having a similarity value higher than 0.76 were found to be only 13. We observed that for very high significance scores the CDF becomes a flat line. In fact, at a threshold of 0.9 or higher, most molecules in a typical diverse database would be singletons with no near neighbors. In the case of USPTO, the average sequence length of a reactant was 28 meaning that the corresponding Tanimoto coefficient values for single (SM) and double (DM) mutations are 0.97 and 0.93. Therefore, if a single or double fragment mutation is found for a molecule, this pair has to be extremely similar to each other. Figure 6 indicates that it is possible to find such pairs if the search DB is large enough like PubChem. In this revision, we refined our approach to avoid ambiguity by avoiding weakly-defined bioactive criteria, or soft thresholds in general. The revised manuscript highlights the more rigorous quantitative hard threshold concept. We added the above analysis in Section 3.3 on page 7 line 5-10 as follows.

To quantify how low the probability of finding such extremely close neighbors of molecules is in a large database, we performed extensive analysis by using AEs as presented in Supplementary Table 4. Considering the cumulative distribution function of AEs obtained with 1.3 million molecules in the USPTO database, only 13 pairs were found to have a T_c value of 0.76 or higher. With a threshold of 0.9 or higher, most molecules in a typical database would be singletons with no near neighbors.

It is worth to notice that AEs provide a powerful high-resolution representation. In our analysis, we considered AEs as the pieces of a molecular jigsaw puzzle. Larger fragments contain the details of smaller fragments. For this reason, the proper choice of the ECFP radius ensures that an AE which is isomorphic to molecule can always be found (*Appendix Figure 1*) (Supplementary Figure 2) However, for a translation task, we find that the optimum ECFP radii should be equal to 1. We added the above point in Section 2.2 on page 4 line 17-23 as follows.

The AE representation does not record any connectivity information. Thus, there is no one-to-one correspondence between molecular structure and the set of AEs. In our analysis, we considered AEs as the pieces of a molecular jigsaw puzzle. Larger pieces (higher radii fragments) encompass small pieces (smaller radii fragments). A proper fingerprint radius ensures that a fragment isomorphic to the molecular structure can be found (Supplementary Figure 2). However, as discussed in section 3, the optimum AE radius for a neural translation task is equal to 1.

2. *"Indeed, the model does not perform particularly well, and the author define a "bioactively similar" concept to increase success by stating that if a predicted reactant is similar to the correct one that's OK. This approach unfortunately reveals a profound misunderstanding of what retrosynthesis is about: one cannot synthesize products from approximate analogs, and the examples in Figure 4 illustrate the problem: there are mistakes in the position and number of aromatic halogens in the predicted reactants."*

It should be kindly noted that we did not define the bioactivity criteria. It has been conventionally used to screen similar molecules in previous studies. Molecules having $T_c \geq 0.85$ tend to exhibit similar biological activities [1–9]. This assumption has been extensively studied, particularly in applications to drug discovery, with different datasets and fingerprints [7, 9–12].

Based on a comparison with various state-of-the-art models, RetroTRAE already outperformed several existing retrosynthetic predictors in terms of the top-1 exact match accuracy. In addition, during revision, we trained RetroTRAE more extensively with data augmentation [13], with and without positional encoding [14], and performed hyperparameter optimization with a new learning rate scheduler [15]. As a result, we achieved an average top-1 exact matching accuracy of 58.3%, including both uni-molecular and bi-molecular reactions.

Unlike SMILES-based methods, multiple degrees of accuracy can be computed by virtue of our model being similarity based. An advantage of using AEs over SMILES is that errors do not lead to invalid or entirely different predictions. Instead, similar predicted sets of AEs lead to molecules that are similar to ground truths. Highly similar predictions can provide useful information for retrosynthesis, complementary to the exact predictions. However, the similarity concept is not a valid measurement tool in assessing the performance of SMILES-character-based models. Therefore, to leverage the native design of our model, we present the results obtained by four different cutoffs, which can be categorized as (a) hard thresholds using the number of tokens and (b) soft thresholds using the Tanimoto coefficient in Table 2, Section 3.3 on page 7. We updated Table 2 in the revised manuscript based on the newest results presented in Appendix Table 3 (Supplementary Table 6).

Based on the above-mentioned points, we added following lines to Section 3.7 on page 11 lines 8-12.

Our approach achieved an average top-1 exact matching accuracy of 58.3%, outperforming existing NMT-based template-free models. The inclusion of single and double fragment mutations, corresponding to 3.3% of the predictions, increased the overall performance of our model to 61.6%, exceeding all current state-of-the-art performance levels, and clearly demonstrated that AEs are useful and informative representation of a molecule.

Also, the following lines are added to Section 2.5 on page 5 lines 28-35.

Unlike SMILES-based methods, small prediction errors of the AE representation do not yield invalid predictions. Thus, multiple degrees of accuracy can be calculated due to the native design of our model. The results were computed with four different cutoffs, which can be categorized as: (a) hard thresholds, and (b) soft thresholds. We define hard thresholds as the discrepancies of one or two fragments. We call arbitrary thresholds based on the Tanimoto coefficient *soft thresholds* such as $T_c \geq 0.85$. These measures are conventionally used to screen similar molecules. For example, molecules having $T_c \geq 0.85$ tend to exhibit similar biological activities [1–9]. This assumption has been tested in multiple studies with different datasets and fingerprints [7, 9–12].

In the revised manuscript, we have placed less emphasis on soft thresholds such as bioactive similarity, and avoided their use when reporting the overall accuracy. Soft thresholds have ambiguity and are difficult to assess objectively due to their length dependence. We were also unable to generalize the characteristics of such similar predictions. By contrast, we placed more emphasis on exact predictions and stricter criteria, SM and DM, to define highly similar predictions. The single and double fragment mutations together account for only 3.3% of the total predictions. They offer several advantages over soft thresholds. These advantages are discussed in Section 2.5 on page 5 line 37-40. We added the below sentences to clarify superiority of hard thresholds.

Hard thresholds offer the following advantages over soft thresholds. First, hard thresholds do not depend on sequence length (Supplementary Table 3). Second, contrary to soft thresholds, they allowed us to easily find the type and number of fragments that deviated from the ground truth. ...

We also agree with reviewer's comment that a desired product cannot be synthesized from approximate analogs. The term 'close analogs' used on page 11 line 18 could be confusing. Thus, to avoid the confusion, the characteristics of single (SM) and double (DM) mutations are further elaborated in the revised manuscript in the first paragraph in Section 3.6, page 8, so as to show that they are significantly superior to structural analogs. For example, atom and connectivity types for the whole structure must be preserved in all SM cases (Figure 4b in revised manuscript). Majority of DM structures also comply with this strict condition. Based on the design principle of AE, predictions with one or two erroneous AEs must correspond to minor variations at peripheral sites. In addition, AEs near the reaction center were robust with respect to hard thresholding. Structural analogs, however, imply differences in certain substructures, functional groups, or several atom types. We believe that such small discrepancies can be easily corrected via visual inspection. To clarify the above in the revised manuscript, we have redrawn the Figure 4 including an example of exact prediction, single and double fragment mutations, and a corresponding performance gain of 3.3%.

In addition to exact predictions, we investigated how much singly and doubly mutated predictions are similar to the ground truth. The first example illustrates an exact prediction (shown in Figure 4a). RetroTRAE predicted 58.1% of the reactions in the test set accurately. The single and double fragment mutations together account for 3.3% of the total predictions. In single mutation cases, atom and connectivity types must be preserved, therefore only two types of structural changes are possible. First, a new environment may appear (or an existing environment may disappear) due to a misplaced single environment (e.g., at the ortho/para/meta position). With this change, all connected atom types must be preserved (Figure 4b). Second, a single existing AE can be added or subtracted at terminal sites. Double mutations are characterized by a misplaced branching AE or a single atom substitution (Figure 4c). If a mutation happens in the middle of a molecule, the AE centered at the mutated site and its direct neighbors are highly likely to be changed, leading to at least three AE mutations.

Based on this observation, SM and DM predictions are much more similar to a ground truth than conventional structural analogs implying differences in certain substructures, functional groups, or several atom types. We believe that these small

discrepancies are easily amendable through a visual comparison with a product. When soft thresholds are used, several AEs can be altered, making the generalization of errors highly difficult.

As we noted earlier the single and double fragment mutations refer to extremely close neighbors of molecules based on the analysis presented in *Appendix Table 1 (Supplementary Table 4)*. We carried out an additional analysis (*Appendix Table 2*)(Supplementary Table 5) to compare two of the commonly used substructural fingerprints in virtual screening (MACCS and RDKit fingerprint) against AEs. We randomly selected 10 singly and 10 doubly mutated predictions and compared mean pair-wise similarities with respect to ground truth and the number of equivalent representations. The mean T_c for AEs was 0.91, while almost none of the mutations were detected by MACCS keys. We found that 17 out of 20 pairs were structurally equivalent. Although not as pronounced as MACCS keys, RDKit fingerprint also yields a mean pair-wise similarity of 0.97. The results of this analyses further support that the predictions obtained by hard thresholds are of exceptional quality. We presented the results in Section 3.4 on page 8 lines 11-17.

To quantify the resolution power of AE in high similarity region, two of the commonly used substructural fingerprints, MACCS and RDKit fingerprint, were compared against AEs (Supplementary Table 5). We randomly selected 10 singly and 10 doubly mutated predictions and compared the mean pair-wise similarities with respect to ground truth and the number of equivalent representations. The mean T_c for AEs was 0.91, while almost none of the mutations were detected by MACCS keys. Seventeen out of 20 pairs were structurally equivalent. The RDKit fingerprint yielded a mean pair-wise similarity of 0.97. These show that the predictions obtained by hard thresholds, SM and DM, are at an exceptional level.

Nevertheless, we cannot generalize the characteristics of the predictions beyond double mutations, albeit within the bounds of bioactive similarity space. The number of variations to consider were nearly impossible to account for. Therefore, we refined our approach to assess the overall performance of our model by avoiding weakly-defined bioactive criteria. The revised manuscript highlights the more rigorous quantitative hard threshold concept instead. We emphasized the above point in Section 3.6, page 11, lines 3-4 as follows.

Nevertheless, we were unable to generalize the characteristics of the predictions beyond DMs, albeit within the bounds of bioactive similarity space.

3. *The work does not represent a significant advance in the field.*

The contribution of our work is not limited to the development of a new prediction method with a better accuracy. We emphasize that this comprehensive study does not only outlines our novel retrosynthesis prediction model based on AEs but also addresses the major limitation of structural fingerprints that precludes their implementations in NLP models. Our work clearly demonstrates that using AE is a new promising approach to develop other generative or sequence-based architectures in addition to conventional SMILES-based approaches. Detailed analysis of

predictions including attention values suggests that models trained with AEs are fully interpretable because each token has a chemical meaning. Therefore, we strongly believe that our approach will be interesting to a broad scope of readers in related fields. To clearly state our conclusions, we rewrote the Conclusion section

... RetroTRAE provides fast and reliable retrosynthetic route planning for substances whose fragmentation patterns are revealed. ... Using AEs has advantages compared with conventional SMILES-based models. First, it need not learn complex grammar of SMILES. Second, each token is an actual substructure of a molecule making a model more interpretable in a chemical sense. Third, no atom mapping procedure is necessary, which can be computationally expensive and introduce additional errors to input data. Detailed analysis of predictions including attention values suggests that models trained with AEs are fully interpretable and AEs with high attention values reveal reaction centers. ... RetroTRAE predicted reactant candidates with an exact match accuracy of 58.3%. In addition to the exact match accuracy, highly similar reactant candidates with single and double mutations were exceptionally similar to ground truth with a p-value $< 10^{-7}$. The overall accuracy with singly and doubly mutated predictions was 61.6%, outperforming current state-of-the-art methods. ... We emphasize that this comprehensive study outlines our novel retrosynthesis prediction model based on AEs and addresses the major limitation of structural fingerprints, which precludes their implementations in NLP models.

Reviewer Point P 1.1 — The authors should be clear in that they are only limiting themselves to the task of single step predictions and not predicting full retrosynthetic pathways. This differentiation is useful for the reader to know the context and applicability of the work carried out.

Reply:

We now clearly mention that our approach operates at the level of AEs predicting transformations from products to reactants in a single-step similar to previous studies [13, 16, 17]. The main reason for focusing on single-step reactions is that the mechanistic descriptions of reactions are not provided in the USPTO database. However, there is no intrinsic limitation for the model to predict multi-step synthetic routes. The model would be able to predict multi-step synthetic routes, when it is combined with a proper search algorithm, such as Monte-Carlo tree search [18, 19], or is run recursively. In its current form, the model can be used in any single-step of a multi-step retrosynthesis [18]. We clarified this aspect particularly in Section 3.5 on page 8 lines 35-41, as well as in other parts of the paper, including the headline of Section 3.2, page 6, line 18.

RetroTRAE operates at the level of AEs predicting transformations from products to reactants in a single-step similar to previous studies [13, 16, 17]. The main reason for focusing single-step reactions is that the mechanistic descriptions of reactions are not provided in the USPTO database. However, there is no intrinsic limitation for the model to predict multi-step synthetic routes. The model would be able to predict multi-step synthetic routes, when it is combined with a proper search algorithm, such as Monte-Carlo

tree search [18, 19]. In its current form, RetroTRAE can be used in any single-step of a multi-step retrosynthesis [18].

Reviewer Point P 1.2 — Abstract: What are the issues of SMILES-based retrosynthesis? This is not clear for a non-expert in the field.

Reply: We have responded to this comment along with our response to Reviewer Point P 1.6, please refer to our answer to P 1.6.

Reviewer Point P 1.3 — Introduction: The authors state that the template-free methods have emerged as means of addressing methodological issues with template-based retrosynthesis. An outline of where the authors think improvements are made should be given, as there are specific regions in which template-based methods overcome methodological challenges, and this is not a general problem.

Reply: We appreciate the reviewer for the helpful comment to improve the quality of our manuscript. To clarify the potential methodological challenges of template-based methods, which can be overcome by template-free methods are described in Introduction section on page 1, from line 42 as follows.

Template-free methods have emerged as an effective means to complement the following issues of template-based methods. Exploring the space of possible reaction templates is challenging because of the vast size of chemical space. If only a limited number of reaction templates are used, template-based methods may not be able to provide novel disconnections [20, 21]. On the contrary, if a large number of reaction templates are considered, computational burden to find a proper template increases significantly. Currently, templates are either hand-crafted by experts [22] or generated from reaction databases with heuristic algorithms [23, 24]. Thus, the degree of template generality/specificity can lead to either low-quality or incomplete recommendations. Lastly, reaction templates are extracted based on atom mapping, which remains a challenging issue for all template-based methods [25]. Atom mapping quality also affects model performance.

Reviewer Point P 1.4 — Line 24, page 3 – The word synthetically accessible should be removed a SMILES is not synthetically accessible and usage of the term can lead to confusion.

Reply: We agree with the reviewer's comment that 'synthetically accessible' can lead to confusion. The phrase 'synthetically accessible' is removed as follows to avoid confusion, and the new sentence is added in the Introduction section, page 2, lines 23-24.

In addition, grammatically valid SMILES are not guaranteed to be semantically valid due to, i.e., explicit valence and kekulization errors.

Reviewer Point P 1.5 — End of page 3 – It has been shown that the transformer learns pairwise relationships between atoms in the reactants and products. In light of this what additional benefit does tokenisation of atomic environments add?

Reply: We deeply appreciate the reviewer's comment, which helped us improve our manuscript. We do agree with the reviewer that the benefits obtained by the usage of AE tokens is one of the key aspects partially overlooked in the current manuscript, and neglected in previous synthetic planning studies in literature. Thus, we tried to emphasize this new aspect more importantly in the revised manuscript, and accordingly added the new 'Model Interpretability' subsection on page 8.

Tokenization is a crucial preprocessing step in many NLP tasks including machine translation which has recently become a widespread tool in chemistry [26]. The main goal of a machine translation task is to generate meaningful sequences using meaningful tokens. From the perspective of a chemist, atom-wise or character-wise tokenization of SMILES, which are commonly used for many NLP-based studies do not produce fully interpretable tokens because many characters in SMILES strings are used to represent the topological characteristics, such as ring-closure or branches. These topological descriptors of SMILES do not correspond to physical entities, atoms and electrons, can be placed in an arbitrary fashion. The advantage of using attention matrices is that it shows the correlation between source and target tokens. 'Interpretability', along with 'explainability' necessitate the existence of meaningful tokens since NLP models tend to learn the relationships between them.

AEs are chemically meaningful tokens in the form of SMARTS patterns as a desired feature. We identified that our model learned the changes in chemical environments around reaction centers in terms of AEs (Figure 3). The model pays more attention to the alterations of AEs (tokens) near the reaction centers because the pairs of tokens with highest attention values correspond to the reaction site or its neighbors. In contrast to our approach, in SMILES-to-SMILES translations, chemical changes mostly occur via "rearrangements of SMILES tokens" due to the conservation of atom types in an ideal reaction dataset. In other words, we don't observe transformations in chemically meaningful tokens, i.e. elements or atom types. Therefore, chemical interpretability and explainability are hampered and remain as an issue in SMILES-based translation.

On the basis of the above summary, we used attention weights to uncover what our model actually learns – or in other words, why our model predicts one retrosynthetic outcome over another. Detailed description of the attention visualizations is provided in Appendix Figure 2, also as a Figure 3 in revised manuscript. The assessment is elaborated in Section 3.5 on page 8 lines 19-34 as follows.

It is often difficult to attribute meaning to the outcomes of deep learning methodologies. We investigated attention weights to uncover what our model actually learns. We identified that our model successfully learned the changes in chemical environments around reaction centers. In contrast to our work, in SMILES-to-SMILES translations chemical changes mostly occur via rearrangements of SMILES tokens rather than actual transformations of chemically meaningful tokens, which hampers chemical interpretability and explainability. To address this issue, Kovács et al. proposed a framework to interpret the results of Molecular Transformer [27].

The attention weight matrices and the fragments with the highest attention values of two example reactions are visualized in Figure 3. The AE that undergoes a change during the reaction has the highest attention value with its changed counterpart. Likewise, the AEs that remain intact tend to have highest attention with itself. The column-wise summations of attention weights indicate the mostly attended AEs of a product by RetroTRAE. To show this, we highlighted the AEs in products that changed during the reactions and their attentions in the reactant side. Indeed, the model pays more attention to altered AEs near the reaction centers as exemplified with ring opening and dissociation reactions. These examples clearly show that AE tokens are chemically meaningful and fully interpretable by themselves as opposed to SMILES tokens.

Reviewer Point P 1.6 — Line 12 page 4 – The problems with SMILES could be clearly outlined in the introduction to aid the readers in understanding why fragment based approaches may have an advantage.

Reply: We appreciate the reviewer for a helpful comment. SMILES-based deep learning NLP models are prone to generate invalid SMILES strings. In a recent study, the top-10 invalidity errors were reported as much as 12.6% [16]. The invalidity errors (SMILES parsing errors) can be attributed to the fragile structure (strong dependence between tokens) of SMILES. A change in a single character often suffices to invalidate an entire SMILES string. The model's interpretability is also hampered by the fact that SMILES tokens are not fully interpretable in a chemical sense. In addition, predicted valid SMILES are not guaranteed to be chemically valid due to explicit valence and kekulization errors. Several attempts have been proposed to ensure syntactic and chemical validity of SMILES predictions [28–30]. In fact, the challenges that SMILES syntax poses prompted the development of alternative syntaxes such as DeepSMILES [31] and SELFIES [32]. We now clearly outlined these aspects in the Introduction section on page 2 line 16-26.

Despite its widespread usage, SMILES easily leads to erroneous predictions because of its fragile and complex grammar. For instance, a single character change is often enough to invalidate an entire SMILES string. Thus, SMILES-based prediction methods tend to make many grammatically invalid predictions reducing their prediction efficiency. In a recent study, the top-10 invalidity error (SMILES parsing errors) was reported as much as 12.6% [16]. To solve this problem, SCROP [28] included a neural-network-based syntax corrector to decrease the invalidity rate. Similarly, other studies [29, 30] focused on determining the causes of invalid SMILES to improve the prediction accuracy. In addition, grammatically valid SMILES are not guaranteed to be semantically valid due to, i.e., explicit valence and kekulization errors. To circumvent these problems, alternative syntaxes such as DeepSMILES [31] and SELFIES [32] were developed.

Reviewer Point P 1.7 — Methods: The authors do not state how they identify uni- and bi- molecular reactions. The reaction datasets contain a multitude of species on the

reactant side of the equation, and not all species are reactive or contribute atoms to the product. How do they identify which ones are relevant from the reaction? Presumably this requires a curation step for which the details are not given.

Reply: We appreciate the reviewer for helpful comments. We used the curated subset of Lowe's grants dataset (USPTO-Full) [33, 34]. The reviewer is indeed correct that not all the species on the reactant side are reactive and contribute to the products. Identification of the agents participating the reaction can only be achieved by atom-mapping. In USPTO-Full, atom maps in the reaction SMILES were derived using Epam's Indigo toolkit. According to Lowe, atom mappings are typically correct, but there are also wrong cases and hence should not be entirely relied on. Jin et al. further refined the USPTO-Full set by removing duplicates and erroneous reactions [21]. This curated dataset contains 480K reactions. Preprocessing steps to remove reagents from reactants are described in refs. [25, 35]. The procedure of separating reagents from the reactants was based on a ">" token in reaction SMILES. By following the same steps, Zheng et al. removed reagents and canonicalized the molecules [28]. Zheng's version presents reactant and product SMILES as well as atom-mapped reaction SMILES. Conclusively, we used this version, which is available in the following repository. <https://github.com/sysu-yanglab/Self-Corrected-Retrosynthetic-Reaction-Predictor/tree/master/data>

The product-reactant pairs were carefully curated. We limited ourselves to single product reactions, corresponding to 97% (465K) of all the available reactions. We then omitted multi-component reactions primarily because they correspond to less than 1.65% of the whole dataset. We set an upper length limit for sequences up to 100 fragments. In this study, we have not used any atom-to-atom mapping algorithm. With forward reactions, we ended up two distinct curated datasets based on the number of reactants, consisting of unimolecular ($R \rightarrow P$) and bimolecular ($R_1 + R_2 \rightarrow P$) type reactions, with a combined size of 414K. Since retrosynthesis implies an abstract backward direction, we named our datasets unimolecular and bimolecular reactions. The Dataset section has now been modified to include the specific points of the curation steps, in Section 2.3 on page 4 lines 38-49.

Preprocessing steps to remove reagents from reactants are described in refs [25, 35], which were based on a ">" token in reaction SMILES. By following this procedure, Zheng et al. provided canonicalized reactant and product SMILES [28]. In addition, there was no reaction class information available in this dataset.

We used Zheng's version of USPTO and carefully curated the product-reactant pairs. We limited ourselves to single product reactions, corresponding to 97% (465K) of all the available reactions. We then omitted multi-component reactions primarily because they occupy less than 1.65% of the whole dataset. We set an upper length limit for sequences up to 100 fragments. In this study, we have not used any atom-to-atom mapping algorithm. With forward reactions, we ended up two distinct curated datasets based on the number of reactants, consisting of unimolecular (\$R \rightarrow P\$ ) and bimolecular (\$R_1 + R_2 \rightarrow P\$ ) type reactions, with a combined size of 414K. Since retrosynthesis implies an abstract backward direction, we named our datasets unimolecular and bimolecular reactions.

Reviewer Point P 1.8 — Methods: the approach is Limited to uni- and bi- molecular data. Can it be extended further to multi component reactions that are prevalent in reaction datasets? If so why have the authors omitted this from the study?

Reply: In this study, we used the USPTO dataset only, because we did not have an access to other proprietary reaction databases. We omitted multi-component reactions primarily because they are observed no more than 1.65% percent of the whole dataset (7906 of 478612 reactions). If there are sufficient number of reaction data for training, RetroTRAE may be able to predict such predictions. We clarified this point in the Dataset section on page 4 lines 43-44.

We then omitted multi-component reactions primarily because they occupy less than 1.65% of the whole dataset.

Reviewer Point P 1.9 — Methods: did the authors separate the validation set prior to training or did they sample the validation set from the training set after training? It is not clear if the validation set was trained on leading to a potential overfitting.

Reply: The validation set was split prior to training and used only for optimizing the hyperparameters. This is now clarified in Section 2.4, page 5, lines 3-4.

The validation set was randomly sampled from the training set (10%) prior to training and used only for optimizing the hyperparameters.

Reviewer Point P 1.10 — Methods: What libraries/packages were used to train the transformer models? What parameter values and configuration settings were used? What tokenization scheme was used?

Reply: We used the open-source chemoinformatics Python module RDKit for obtaining the SMARTS patterns of AEs and the PyTorch machine learning library for constructing and training the model. The model configuration was set similar to the original Transformer paper, except few parameters. The normalization layer was applied prior to self attention, multi-head attention and feed-forward operations, respectively. We also normalized the outputs of the encoder and decoder. Word-wise tokenization was applied by using the SentencePiece tokenizer [36]. The details of our key hyperparameters are described in the Supplementary Information. For clarification, we have added a new paragraph at the end of the Training Details section.

The open-source RDKit [37] module version 2020.03.1 was utilized to generate ECFPs and AEs. The PyTorch [38] machine learning library was used for constructing and training the model. The model was configured similarly to the original Transformer paper, except the normalization layer was applied prior to self-attention, multi-head attention and feed-forward operations, respectively. The outputs of the encoder and decoder were also normalized. Word-wise tokenization was applied by using the SentencePiece tokenizer [36]. The details of our key hyperparameters and hyperparameter space are described in

Supplementary Table 1.

Reviewer Point P 1.11 — The authors state that non-transformer models do not perform as well as transformer based models for the task of retrosynthesis. Is there any evidence to support this beyond the top1 accuracy that is not a suitable metric for assessing the task of retrosynthesis? As there are many possible disconnections that lead to the product (i.e. many reactions that can be used) top 1 accuracy fails to account for the other viable disconnections.

Reply: First of all, we should have been more careful about our statement. Though the statement was based on the numerical results presented in Table 3, we needed to describe the results more meticulously while generalizing. There was no theoretical justification for the use of this statement since we have not compared each and every retrosynthetic tool available in the literature. As the reviewer pointed out elsewhere, the absence of the retrosynthetic planning tool AiZynthfinder can be given as an obvious example. Therefore, we now deleted the argument but simply stated that the Transformer-based models have shown promising results.

Top-N accuracy is the most commonly used evaluation metric assessing retrosynthetic model performances. It is particularly a legitimate measure of assessing a single-step retrosynthesis predictor trained with the USPTO dataset owing to the fact that only 6% of the USPTO have at least two sets of reactants [33, 34, 39]. The usage of top-N accuracy for the evaluation of retrosynthesis prediction has recently been disputed because, with each prediction, the model tends to find the next frequently observed answer among reaction dataset rather than making a chemically more meaningful prediction [17, 39]. The rationale behind this criticism is in line with our principle of reporting top-1 performance of the model exclusively, rather than top-10, top-20, or even top-50. A few alternative metrics are newly suggested such as Round-trip [17], and MaxFrag [13]. Furthermore, top-1 accuracy is comparable to the process of counting the number of successfully solved routes (if all predictions are in stock database), as another way of assessing the retrosynthetic model performance. To clarify this point, we now have added the above discussion to the end of the Evaluation section with additional references.

However, the analyses showed that only 6% of the USPTO dataset has at least two sets of reactants [33, 34, 39]. Thus, using top-1 accuracy is a legitimate measure to assess a single-step retrosynthesis predictor trained on the USPTO dataset. Top-N accuracy for evaluating retrosynthesis prediction has recently been disputed because, with each prediction, a model tends to find the next frequently observed answer among reactions in a dataset rather than making a chemically more meaningful prediction [17, 39]. A few alternative metrics were newly suggested, such as Round-trip [17], and MaxFrag [13].

Reviewer Point P 1.12 — It's not possible to discuss retrosynthesis without citing the Waller 2018 Nature paper, the AiZynthfinder of Genheden et al., and the Chematika/Synthia program and many applications by Grzybowski and coworkers (even if not AI based).

The influence of datasets on retrosynthesis planning performance (Thakkar et al.) must also be cited and taken into account.

Reply: We appreciate the reviewer for pointing out that template-based approaches are not properly covered in the current manuscript. We do agree with the reviewer's suggestion, indicating the importance of template-based approaches. To address this issue, we added a new paragraph presenting template-based approaches to the Introduction section on page 1 lines 35-41. The recommended references are cited properly.

Template-based approaches use reaction templates to predict reactants from a product. Reaction templates are extracted from data using algorithms or encoded manually. For manual encoding, deep chemical expertise and management of complex transformation rules are needed [20, 22, 40, 41]. Data-driven approaches, however, enabled automated extraction of large reaction templates from reaction data [18-20, 23, 24, 42, 43]. For retrosynthesis prediction, each template is applied to a product to find a match, subgraph isomorphism. If a proper isomorphism is found, a product is transformed depending on the template. This process continues until chemically reasonable pathways are identified [19].

Reviewer Point P 1.13 — The claim that transformers are better than template-based is entirely unfounded and not supported by the analysis in Table 3. In Table 3, in the comparison of transformer-based retrosynthesis, please a comparison with the model of ref. 15 Schwaller et al. and with the AiZynthfinder must be included.

Reply: We acknowledge that our statement was not clear enough. We did not intend to compare the Transformer-based methods with all the other methods including template-based methods. Our purpose was to compare the Transformer-based methods with previous template-free methods that used other approaches than Transformer.

The retrosynthetic performance of the Molecular Transformer (MT) architecture developed by Schwaller for forward-reaction prediction was actually given in Table 3. The same Molecular Transformer architecture together with the same molecular representation is well-adapted for retrosynthesis by Zheng et al. (MT + syntax corrector), Tetko et al. (MT + augmentation), and Lin et al. (MT + different SMILES tokenization). Those works are the successful extensions of Schwaller's method, which was designed for forward-reaction prediction, and are presented in Table 3. Later, as mentioned briefly in our reply to Reviewer Point P 1.11, Schwaller et al. disputed the commonly used top-N accuracy metric. As they described in their report, they evaluated the retrosynthetic performance of the Molecular Transformer architecture on four different metrics that are proposed for the first time by themselves. These new metrics also require a forward-reaction predictor. Thus, we believe that these newly introduced metrics are outside the scope of our study.

For a comparison with top performing template-based models, we listed the top-1 accuracy of the neural engine of AiZynthfinder [19], reported as a range of 43-72% for the filtered USPTO dataset depending on the sizes of template libraries that were used to train template prioritization models [39]. We also reproduced the success rate of AiZynthFinder of 55.7% for 5000 ChEMBL compounds (originally tested

over a data set of 100 ChEMBL compounds, and 55 routes are found). We further added the top-1 accuracy of 50.1% reported by Segler and Waller using Reaxys [39]. However, it should be noted that each template-based model uses different datasets and template extraction methods, which can heavily affect the performance of a model. We have updated Table 3 by including the results of AiZynthfinder, and included corresponding discussions in Section 3.7, page 11, lines 17-22.

For a comparison with top performing template-based models, we listed the top-1 accuracy of AiZynthfinder [19]. The accuracy was reported as a range of 43-72% on the filtered USPTO dataset depending on the sizes of template libraries that were used to train template prioritization models [39]. Segler and Waller reported a top-1 accuracy of 50.1% using Reaxys [20]. It should be noted that each template-based model used different training/test datasets and template extraction methods, which affect model's performance.

Reviewer Point P 1.14 — The format of all references is completely wrong.

Reply: References are corrected and now compatible with the guideline.

Reviewer 2

The article deals with important research directions as retrosynthesis prediction; it is well written and structured. There is also a live GitHub repository supporting the work. However, I have two questions for the authors that should be answered before the final publication.

Reply: We thank the reviewer for his/her positive comments. In this revision, we have followed the suggestions made by the reviewer and updated the manuscript accordingly.

Reviewer Point P 2.1 — If a molecule tokenized with Atomic Environments instead of SMILES letters is fed to an NLP model, there must be an order and a rule to construct this order from single AEs. There are places in the article where the authors speak about sets, but the final model also has the positional encoding layer; therefore, the position is valuable. The model should have the same problems that other scientists are trying to prevent with canonical/augmented SMILES. The authors should undoubtedly clarify this issue.

Reply: We thank the reviewer for pointing this out. AEs were ordered in the same order as the bit-vectors generated by the hashed-Morgan fingerprint algorithm implemented in RDKit. Our code converts molecules into AEs in this default order. However, because there is no connectivity information between individual AEs, the order does not matter. This also conforms with attention mechanism, which is a permutation-invariant operation. We have used the positional encoding because it is preferable over non-positional encoding case according to a previous study [14]. To test this claim, we trained the

model from scratch without positional encoding. The results indicate that the models with positional encoding perform better than those trained on without imposing position information (*Appendix Table 3* (Supplementary Table 6)). This is consistent with the observation by Jaegle et al. [14].

In addition, during revision, we trained RetroTRAE with data augmentation [13], and performed hyperparameter optimization with a new learning rate scheduler [15]. Augmentation slightly improved the results and stabilized the model's learning because more data and randomness were added to the network [13]. RetroTRAE achieved 56.4% and 60.1% accuracy in predicting exact matches over augmented (x10) uni- and bi-molecular datasets, respectively. As a result, we achieved an average top-1 exact matching accuracy of 58.3%, including both uni-molecular and bi-molecular reactions. All of the above results are presented in *Appendix Table 3* (Supplementary Table 6). We updated the Table 2 in revised manuscript accordingly. We presented above points in the first paragraph of Section 3.3

RetroTRAE has reached top-1 exact match accuracies of 56.4% and 60.1% trained with 10 times augmented uni- and bi-molecular datasets. Augmentation slightly improved the results and stabilized the model's learning since more data and randomness were added to the network [13]. Although the AE representation is permutation invariant, the models with positional encoding perform better than those trained on without using positional information (Supplementary Table 6). This is consistent with the observation by Jaegle et al. [14].

Reviewer Point P2.2 — Biosimilarity brings only higher values in the tables and nothing more. It is known that 0.85 works for biosimilarity. However, chemists face many more simple reactants (they might be structurally similar, but the similarity coefficient maybe even be < 0.60). In a general model for retrosynthesis, such biased criteria should be avoided.

Reply: We again thank the reviewer for this critical comment that indeed improved the quality of this manuscript. Following a more careful analysis, we avoided using the biosimilarity criteria for reporting the overall model accuracy. We placed less emphasis on soft thresholds such as bioactive similarity in the revised manuscript based on the set of arguments presented below.

Unlike SMILES-based methods, multiple degree of accuracy can be computed by virtue of our model being similarity based. The similarity concept is not a valid measurement tool in performance assessment for SMILES-character-based models. An advantage of using AEs over SMILES is that errors do not lead to invalid or entirely different predictions, instead, they lead to similar molecules to ground truths. Highly similar predictions can provide useful information for retrosynthesis, complementing the exact predictions. Therefore, to leverage the native design of our model, we have presented the results obtained by four different cutoffs, can be categorized as (a) hard thresholds, and (b) soft thresholds in Table 2, Section 3.3. We added the following sentences to Section 3.3 on page 7 lines 1-5.

One of the advantages of using AEs over SMILES is that a few errors do not lead to invalid predictions. Thus, we investigated how much the success rate can be improved by easing the threshold without losing functionality of the retrosynthetic framework. When single mutations (SM) were allowed, the success rates of uni-molecular and bi-molecular

reactions increased to 58.1% and 60.9%, respectively. The corresponding numbers for double mutations (DM) were 60.5% and 62.7%.

By contrast, we are in full agreement with the reviewer that soft thresholds are somewhat ill-defined and difficult to assess objectively, though they are conventionally used to screen similar molecules. We were unable to generalize the characteristics of the predictions beyond double mutations, albeit within the bounds of bioactive similarity space. The number of variations to consider were nearly impossible to account for. We emphasized the above point in Section 3.6, page 11, line 3-4.

We refined our approach of assessing overall model performance by avoiding weakly-defined bioactive criteria. We placed more emphasis on exact predictions and used stricter criteria, single and double fragment mutations, to define highly similar predictions. The revised manuscript highlights the more rigorous quantitative hard threshold concept. The single and double fragment mutations together account for only 3.3% of the total predictions. Hard thresholds offer several advantages over soft thresholds. These advantages are given in Section 2.5 on page 5 lines 36-38. We added the discussion to clarify the superiority of hard thresholds.

Hard thresholds offer the following advantages over soft thresholds. First, hard thresholds do not depend on sequence length (Supplementary Table 3). Second, contrary to soft thresholds, they allowed us to easily find the type and number of fragments that deviated from the ground truth. ...

The characteristics of single (SM) and double (DM) mutations are further elaborated in the revised manuscript in the first paragraph in Section 3.6, page 8, to show that they are far superior to structural analogs. For example, the atom and connectivity types for the whole structure must be preserved in all SM cases (Figure 4b in revised manuscript). Majority of DM structures also comply with this strict condition. Based on the design principle of AE, predictions with one or two erroneous AEs must correspond to minor variations at peripheral sites. In addition, AEs near the reaction center were robust with respect to hard thresholding. Structural analogs, however, imply differences in certain substructures, functional groups, or several atom types. We believe that such small discrepancies can be easily corrected through visual inspection. To clarify the above in the revised manuscript, we have provided a new version of Figure 4 including an example of exact prediction, single and double fragment mutations, and corresponding performance gain as 3.3%.

In addition to exact predictions, we investigated how much singly and doubly mutated predictions are similar to the ground truth. The first example illustrates an exact prediction (shown in Figure 4a). RetroTRAE predicted 58.1% of the reactions in the test set accurately. The single and double fragment mutations together account for 3.3% of the total predictions. In single mutation cases, atom and connectivity types must be preserved, therefore only two types of structural changes are possible. First, a new environment may appear (or an existing environment may disappear) due to a misplaced single environment (e.g., at the ortho/para/meta position). With this change, all connected atom types must be preserved (Figure 4b). Second, a single existing AE can be added or subtracted at terminal sites. Double mutations are characterized by a misplaced branching AE or a single atom substitution (Figure 4c). If a mutation happens in the middle of a molecule, the AE centered at the mutated site and its direct neighbors

are highly likely to be changed, leading to at least three AE mutations.

Based on this observation, SM and DM predictions are much more similar to a ground truth than conventional structural analogs implying differences in certain substructures, functional groups, or several atom types. We believe that these small discrepancies are easily amendable through a visual comparison with a product. When soft thresholds are used, several AEs can be altered, making the generalization of errors highly difficult.

The single and double fragment mutations refer to extremely close neighbors of molecules. As shown in Section 3.9, Figure 6, the probability of finding such extremely close neighbors of molecules is extremely low even in the PubChem DB with 111 million molecules. To be precise, we performed additional analyses by using AEs as presented in *Appendix Table 1 (Supplementary Table 4)*. Considering the cumulative distribution function of AEs obtained by using 1.3 million molecules in USPTO database, only 13 out of 20 pairs were found to have a similarity value higher than 0.76. In fact, at a threshold of approximately 0.9, most molecules in a typical diverse database would be singletons with no near neighbors. We have discussed this in Section 3.3, page 7, lines 5-10.

To quantify how low the probability of finding such extremely close neighbors of molecules is in a large database, we performed extensive analysis by using AEs as presented in *Supplementary Table 4*. Considering the cumulative distribution function of AEs obtained with 1.3 million molecules in the USPTO database, only 13 pairs were found to have a T_c value of 0.76 or higher. With a threshold of 0.9 or higher, most molecules in a typical database would be singletons with no near neighbors.

Another comparison is presented in *Appendix Table 2 ((Supplementary Table 5)* Two of the commonly used substructural fingerprints in virtual screening (MACCS and RDKit fingerprint) were compared against AEs. We randomly selected 10 SM and 10 DM pairs and compared their mean pair-wise similarity and the number of equivalent representations. The mean T_c for AEs was 0.91, however, almost none of the changes in AEs were detected by MACCS keys. We found 17 out of 20 pairs structurally equivalent. Although not as pronounced as MACCS keys, RDKit fingerprint yields a mean pair-wise similarity of 0.97. The results of this analyses further support that the predictions obtained by hard thresholds are of exceptional quality. We present the results in Section 3.4, page 8, lines 10-17.

To quantify the resolution power of AE in high similarity region, two of the commonly used substructural fingerprints, MACCS and RDKit fingerprint, were compared against AEs (*Supplementary Table 5*). We randomly selected 10 singly and 10 doubly mutated predictions and compared the mean pair-wise similarities with respect to ground truth and the number of equivalent representations. The mean T_c for AEs was 0.91, while almost none of the mutations were detected by MACCS keys. Seventeen out of 20 pairs were structurally equivalent. The RDKit fingerprint yielded a mean pair-wise similarity of 0.97. These show that the predictions obtained by hard thresholds, SM and DM, are at an exceptional level.

Reviewer 3

The authors are proposing to use atomic environments for transformer based synthesis prediction. This is an interesting idea that I believe has been underexploited. However, I believe the current version needs to be significantly improved to be publishable.

Reviewer Point P 3.1 — Not only the datasets but the code needs to be open sourced so the results can be reproducible.

Reply: We appreciate the reviewer for pointing the important point. We uploaded all the dataset and the source code for training and test.

Reviewer Point P 3.2 — Most references are wrongly formatted

Reply: References are formatted as per guidelines.

Reviewer Point P 3.3 — Important work by Alpha Lee et al related to the quality of the USPTO and interpretability of chemical reaction is ignored see for instance published earlier this year in Nature Communications <https://www.nature.com/articles/s41467-021-21895-w>

Reply: We agree with the reviewer that the model interpretability is overlooked in the current manuscript, and mostly neglected in previous synthetic planning studies in literature. We thus tried to emphasize more on this new aspect more importantly in the revised paper, and accordingly added a new 'Model Interpretability' subsection in page 8.

Tokenization is a crucial preprocessing step in many NLP tasks including machine translation which has recently become a widespread tool in chemistry [26]. The main goal of a machine translation task is to generate meaningful sequences from meaningful tokens. From a chemist perspective, atom-wise or character-wise tokenization of SMILES, which are commonly used for many NLP-based studies do not produce fully interpretable tokens because many characters in SMILES strings are used to represent the topological characteristics, such as ring-closure or branches. These topological descriptors of SMILES do not correspond to physical atoms and can be placed in an arbitrary fashion.

The advantage of using attention matrices is that it shows the correlation between source and target tokens. 'Interpretability', along with 'explainability' necessitate the existence of meaningful tokens since NLP models tend to learn the relationships between them. AEs are chemically meaningful tokens in the form of SMARTS patterns as a desired feature. The algorithm converts molecular structures to the fully interpretable set of AEs suitable for NLP applications. We identified that our model learned the changes in chemical environments around reaction centers in terms of AEs. The model pays more attention to the alterations of AEs (tokens) near the reaction centers because the token pairs with the highest attention values correspond to the reaction site or its neighbors. In contrast to AEs, in SMILES-to-SMILES translations, chemical changes mostly occur via "rearrangements of SMILES tokens" due to the conservation of atom types in an ideal reaction dataset. In other words, we don't observe transformations

in chemically meaningful tokens, i.e, elements or atom types. Therefore, chemical interpretability and explainability are hampered and remain as an issue in SMILES-based translation.

On the basis of the above summary, we used attention weights to uncover what our model actually learns – or in other words, why our model predicts one retrosynthetic outcome over another. Detailed description of the attention visualizations is provided in Appendix Figure 2, also as Figure 3 in the revised manuscript. The assessment is elaborated in Section 3.5 on page 8 lines 19-34 as follows.

It is often difficult to attribute meaning to the outcomes of deep learning methodologies. We investigated attention weights to uncover what our model actually learns. We identified that our model successfully learned the changes in chemical environments around reaction centers. In contrast to our work, in SMILES-to-SMILES translations chemical changes mostly occur via rearrangements of SMILES tokens rather than actual transformations of chemically meaningful tokens, which hampers chemical interpretability and explainability. To address this issue, Kovács et al. proposed a framework to interpret the results of Molecular Transformer [27].

The attention weight matrices and the fragments with the highest attention values of two example reactions are visualized in Figure 3. An AE that undergoes a change during the reaction has the highest attention value with its changed counterpart. Likewise, the AEs that remain intact tend to have highest attention with itself. The column-wise summations of attention weights indicate the mostly attended AEs of a product by RetroTRAE. To show this, we highlighted the AEs in products that changed during the reactions and their attentions in the reactant side. Indeed, the model pays more attention to altered AEs near the reaction centers as exemplified with ring opening and dissociation reactions. These examples clearly show that AE tokens are chemically meaningful and fully interpretable by themselves as opposed to SMILES tokens.

Reviewer Point P3.4 — The Bioactivity similarity idea doesn't make any sense. The cut-off chosen, here set to 0.85 is highly dependent on which fingerprint is used. And in practice one wants to synthesize a specific compound not a similar compound that might already have been synthesized or is not interesting due to various reasons.

Reply: The reviewer indeed raises an important question about the fingerprint dependency of similarity cutoff values. As the reviewer correctly pointed out, similarity cut-offs indicate different meanings in different fingerprints. The conventional way of selecting similarity thresholds, which we called "soft (arbitrary) thresholds" in the manuscript, are fingerprint dependent. Although soft thresholds are ill-defined and difficult to assess objectively, they have been frequently used to screen similar molecules. For example, pairs of molecules having $T_c \geq 0.85$ tend to exhibit similar biological activities [1–9]. This assumption has been tested in multiple studies with different datasets and fingerprints [7, 9–12]. Unfortunately, it is unlikely to rationalize a specific similarity value as an activity indicator for various fingerprints.

By contrast, AE formalism offers a powerful high-resolution representation, which is advantageous over other forms of fingerprints. This feature is particularly useful in the context of fingerprint dependency of soft thresholds. To demonstrate, we generated the similarity value distributions of various structural fingerprints available in RDKit using 1.3 million molecules in the USPTO dataset. *Figure 3* in the appendix (also in Supplementary Figure 3) allows us to compare similarity values of a set of fingerprints against each other. For instance, within a more reliable region bounded by a p-value greater than 0.01 (equivalent to $T_c \leq 0.32$ with unified AEs), Avalon, MACCS keys, RDKit and Atom pairs fingerprints all yielded higher T_c values. Topological torsion is the only exception and yielded slightly lower similarity values than AEs. The region beyond the p-value 0.01 is grayed out since the curves are expected to be less reliable. CDF becomes a flat line for very high significance scores. Here, the similarity cut-off of AEs of $T_c = 0.85$ corresponds to a very low p-value of 6×10^{-6} . These results indicate that a chosen cutoff based on AEs lies at a lower similarity level relative to the other fingerprints. We have provided the above analysis in the first paragraph of the Section 3.4, page 8.

AE formalism offers a higher resolution power than other fingerprints. This feature is particularly useful in terms of the context of fingerprint dependency of soft thresholds, Tanimoto coefficient. To demonstrate, we generated the similarity value distributions of various structural fingerprints available in RDKit using 1.3 million molecules in the USPTO dataset (Figure 3). For instance, within a region where a p-value is greater than 0.01 (equivalent to $T_c \leq 0.32$ with unified AEs), Avalon, MACCS keys, RDKit and Atom pairs fingerprints all yielded higher T_c values. Topological torsion was the only exception and yielded slightly lower similarity values than AEs. These results indicate that chosen cutoffs based on AEs lie at a lower similarity level and statistically more significant than other fingerprints.

In addition to the above discussion on similarity-fingerprint dependency, similarity cutoffs have been thoroughly reconsidered, re-examined, and we now avoided using the biosimilarity criteria when reporting the overall model accuracy. Further clarification and explanation of the rationale are given below.

Unlike SMILES-based methods, multiple degree of accuracy can be computed by virtue of our model being similarity based. The similarity concept is not a valid measurement tool in performance assessment for SMILES-character-based models. An advantage of using AEs over SMILES is that errors do not lead to invalid or entirely different predictions, instead, lead to similar molecules to ground truths. Highly similar predictions can provide useful information for retrosynthesis, complementary to the exact predictions. Therefore, to take full advantage of the native design of our model, we have presented the results obtained by four different cutoffs, falling into two categories: (a) hard thresholds, and (b) soft thresholds in Table 2, Section 3.3. We added the following sentences to Section 3.3 on page 7 lines 1-5.

One of the advantages of using AEs over SMILES is that a few errors do not lead to invalid predictions. Thus, we investigated how much the success rate can be improved by easing the threshold without losing functionality of the retrosynthetic framework. When single mutations (SM) were allowed, the success rates of uni-molecular and bi-molecular reactions increased to 58.1% and 60.9%, respectively. The corresponding numbers for double mutations (DM) were 60.5% and 62.7%.

By contrast, we are in full agreement with the reviewer that soft thresholds are somewhat ill-defined and difficult to assess objectively. We were unable to generalize the characteristics of the predictions beyond double mutations, albeit within the bounds of bioactive similarity space. The number of variations to consider were nearly impossible to account for. We emphasized the above point in Section 3.6, page 11, line 3-4.

We refined our approach to assess overall model performance by avoiding weakly-defined bioactive criteria. We placed more emphasis on exact predictions and used stricter criteria, single and double mutations, for highly similar predictions. The revised manuscript highlights the more rigorous quantitative hard threshold concept. The single and double fragment mutations together account for only 3.3% of the total predictions. Hard thresholds offer several advantages over soft thresholds. These advantages are given in Section 2.5 on page 5 lines 36-38. We added the corresponding discussion to clarify the superiority of hard thresholds.

Hard thresholds offer the following advantages over soft thresholds. First, hard thresholds do not depend on sequence length (Supplementary Table 3). Second, contrary to soft thresholds, they allowed us to easily find the type and number of fragments that deviated from the ground truth. ...

The characteristics of single and double fragment mutations are further elaborated in the revised manuscript at the first paragraph in Section 3.6, page 8, so as to show that they are far superior to structural analogs. For example, atom and connectivity types for the whole structure must be preserved in all SM cases (Figure 4b in revised manuscript). Majority of DM structures also comply with this strict condition. Based on the design principle of AE, predictions with one or two erroneous AEs must correspond to minor variations at peripheral sites. In addition, AEs near the reaction center were robust with respect to hard thresholding. Structural analogs, however, imply differences in certain substructures, functional groups, or several atom types. We believe that such small discrepancies can be easily corrected by experienced chemists. To clarify the above in the revised manuscript, we redrew the Figure 4 including an example of exact prediction, single and double fragment mutations, and corresponding performance gain of 3.3%.

In addition to exact predictions, we investigated how much singly and doubly mutated predictions are similar to the ground truth. The first example illustrates an exact prediction (shown in Figure 4a). RetroTRAE predicted 58.1% of the reactions in the test set accurately. The single and double fragment mutations together account for 3.3% of the total predictions. In single mutation cases, atom and connectivity types must be preserved, therefore only two types of structural changes are possible. First, a new environment may appear (or an existing environment may disappear) due to a misplaced single environment (e.g., at the ortho/para/meta position). With this change, all connected atom types must be preserved (Figure 4b). Second, a single existing AE can be added or subtracted at terminal sites. Double mutations are characterized by a misplaced branching AE or a single atom substitution (Figure 4c). If a mutation happens in the middle of a molecule, the AE centered at the mutated site and its direct neighbors are highly likely to be changed, leading to at least three AE mutations.

Based on this observation, SM and DM predictions are much more similar to a ground truth than conventional structural analogs implying differences in certain substructures, functional groups, or several atom types. We believe that these small discrepancies are

easily amendable through a visual comparison with a product. When soft thresholds are used, several AEs can be altered, making the generalization of errors highly difficult.

The single and double fragment mutations refer to extremely close neighbors of molecules. As shown in Section 3.9, Figure 6, the probability of finding such extremely close neighbors of molecules is extremely low in the PubChem DB with a size of 111 million molecules. To be exact, we performed additional analysis by using AEs as presented in *Appendix Table 1 (Supplementary Table 4)*. Considering the cumulative distribution function of AEs obtained by using 1.3 million molecules in USPTO database, the number of pairs having a similarity value higher than 0.76 were found to be only 13. In fact, at a threshold of 0.9 or higher, most molecules in a typical diverse database would be singletons with no near neighbors. We added this analysis in Section 3.3, on page 7 line 5-10.

To quantify how low the probability of finding such extremely close neighbors of molecules is in a large database, we performed extensive analysis by using AEs as presented in *Supplementary Table 4*. Considering the cumulative distribution function of AEs obtained with 1.3 million molecules in the USPTO database, only 13 pairs were found to have a T_c value of 0.76 or higher. With a threshold of 0.9 or higher, most molecules in a typical database would be singletons with no near neighbors.

Another comparison is presented in *Appendix Table 2 (Supplementary Table 5)*. Two of the commonly used substructural fingerprints in virtual screening (MACCS and RDKit fingerprint) were compared against AEs. We randomly selected 10 single mutation and 10 double mutation pairs and compared pair-wise mean similarity as well as the number of equivalent representations. The mean T_c for AEs was 0.91, however, almost none of the changes in AEs were detected by MACCS keys. We found that 17 out of 20 pairs were structurally equivalent. Although not as pronounced as MACCS keys, RDKit fingerprint yields a pair-wise mean similarity of 0.97. The results of this analyses further support that the predictions obtained by hard thresholds are of exceptional quality. We presented the results in Section 3.4, page 8, lines 10-17.

To quantify the resolution power of AE in high similarity region, two of the commonly used substructural fingerprints, MACCS and RDKit fingerprint, were compared against AEs (*Supplementary Table 5*). We randomly selected 10 singly and 10 doubly mutated predictions and compared the mean pair-wise similarities with respect to ground truth and the number of equivalent representations. The mean T_c for AEs was 0.91, while almost none of the mutations were detected by MACCS keys. Seventeen out of 20 pairs were structurally equivalent. The RDKit fingerprint yielded a mean pair-wise similarity of 0.97. These show that the predictions obtained by hard thresholds, SM and DM, are at an exceptional level.

1 Appendix

SMILES : O=C1CCCC(=O)O1

Picture created by SMARTSviewer [smartsview.zbh.uni-hamburg.de]

SMARTS : [O;R;D2]1-[C;R;D3](=[O;!R;D1])-[CH2;R;D2]-[CH2;R;D2]-[CH2;R;D2]-[C;R;D3]-1=[O;!R;D1]

The above SMARTS is belong to ECFP radius = 3, central atom is the oxygen atom in the ring.

Legends

Figure 1: Atom environments

Table 1: Probability of finding pairs of molecules above the given thresholds. The results are based on the CDF generated by using 1.3 million compounds using AEO + AE2.

Thresholds	10%	1%	0.1%	0.01%	0.001%
Tanimoto metric	>.24	>.33	>.42	>.53	>.76

Table 2: Similarity score comparison of MACCS, RDKit and AE representations within the high similarity regime tested on single and double mutated predictions of RetroTRAE.

Fingerprint type	T_c of SM	T_c of DM	$T_c = 1.0$	Average
MACCS	0.99	0.99	17	0.99
RDKit	0.99	0.95	3	0.97
AEs	0.94	0.88	0	0.91

Table 3: Results of data augmentation (x10) and positional encoding trained with Karpov’s cyclic learning scheduler strategy.

	Unimolecular			Bimolecular		
	$T_c = 1.0$	$T_c \geq 0.85$	\bar{T}_c	$T_c = 1.0$	$T_c \geq 0.85$	\bar{T}_c
Positional encoding	55.4	68.1	88	58.3	63.4	77
No Positional encoding	54.2	67.2	87	56.9	62.1	76
x10 Aug (products only)	56.4	68.2	88	60.1	64.3	79
x10 Aug (products+reactants)	44.1	64.0	84	-	-	-

1) Unimolecular reaction example (ring-opening)

2) Bimolecular reaction example (dissociation)

Figure 2: Visualisation of decoder attention and interpretability of RetroTRAЕ are exemplified in unimolecular (upper) and bimolecular (lower) reactions. (Left) Attention weight matrices and column-wise attention sums are displayed. Product AEs with high attention values correspond to the reaction centers of the products. (Right) Highly correlated AE pairs between reactant and product sequences are visualized. The widths of connections are proportional to attention values. The AE pairs with highest attention values correspond to the reaction centers and disconnection sites. The altered AEs surrounding the reaction center with high attention scores are highlighted.

Figure 3: The statistical equivalences between the similarity scores of various structural fingerprints. The region beyond the p-value 0.01 is grayed out since the curves are expected to be less reliable. The similarity cut-off of $Tc = 0.85$ corresponds to a very low p-value of 6×10^{-6} .

1.0.1 Raw data to generate Table 2

Ten single mutant (SM) examples.

- ['Product : O=C(NOCc1ccccc1)c1cc(F)c(F)cc1Nc1cccc(F)c1', 'Reactant1 : NOCc1ccccc1', 'Reactant2 : O=C(O)c1cc(F)c(F)cc1Nc1cccc(F)c1', 'Prediction2 : O=C(O)c1cc(F)ccc1Nc1ccc(F)c(F)c1', {'maccs': 1.0, 'rdkit4': 1.0, 'aes': 0.95}],
- ['Product : CC(C)(C)OC(=O)N1Cc2cc(NC(=O)c3ccccc3NCc3ccc(F)cc3)ccc2C(C)(C)Cl', 'Reactant1 : CC(C)(C)OC(=O)N1Cc2cc(N)ccc2C(C)(C)Cl', 'Reactant2 : O=C(O)c1ccccc1NCc1ccc(F)cc1', 'Prediction2 : O=C(O)c1ccccc1NCc1cccc(F)c1', {'maccs': 1.0, 'rdkit4': 0.9787234042553191, 'aes': 0.9523809523809523}],
- ['Product : Cc1c(Cl)nn(C)c1-c1esc(C(=O)NC(Cc2cccc(F)c2)CN2C(=O)c3ccccc3C2=O)c1', 'Reactant1 : Cc1c(Cl)nn(C)c1-c1esc(C(=O)O)c1', 'Reactant2 : NC(Cc1cccc(F)c1)CN1C(=O)c2ccccc2C1=O', 'Prediction2 : NC(Cc1ccc(F)cc1)CN1C(=O)c2ccccc2C1=O', {'maccs': 1.0, 'rdkit4': 0.9838709677419355, 'aes': 0.9545454545454546}],
- ['Product : CC(=O)Nc1cccc2cc(OCc3ccccc3)ccc12', 'Reactant1 : BrCc1ccccc1', 'Reactant2 : CC(=O)Nc1cccc2cc(O)ccc12', 'Prediction2 : CC(=O)Nc1cc(O)cc2cc(O)ccc12', {'maccs': 0.9, 'rdkit4': 0.9666666666666667, 'aes': 0.9473684210526315}],
- ['Product : COc1cc2nccc(Oc3ccc(C(=O)c4ccccc4C(F)(F)F)cc3)c2cc1OC', 'Reactant1 : COc1cc2nccc(Cl)c2cc1OC', 'Reactant2 : O=C(c1ccc(O)cc1)c1ccccc1C(F)(F)F', 'Prediction2 : O=C(c1ccccc1)c1ccc(O)cc1C(F)(F)F', {'maccs': 1.0, 'rdkit4': 0.9743589743589743, 'aes': 0.9473684210526315}],
- ['Product : CC(C)C(=O)Nc1cccc(C2CCN(Cc3ccc(Oc4ccc(Cl)c(Cl)c4)cc3)CC2)c1', 'Reactant1 : CC(C)C(=O)Nc1cccc(C2CCNCC2)c1', 'Reactant2 : O=Cc1ccc(Oc2ccc(Cl)c(Cl)c2)cc1', 'Prediction2 : O=Cc1cccc(Oc2ccc(Cl)c(Cl)c2)c1', {'maccs': 1.0, 'rdkit4': 0.967741935483871, 'aes': 0.9375}],
- ['Product : Cc1sc(C(=O)NC(Cc2cccc(F)c2)CN2C(=O)c3ccccc3C2=O)cc1-c1c(Br)cn1C', 'Reactant1 : Cc1sc(C(=O)O)cc1-c1c(Br)cn1C', 'Reactant2 : NC(Cc1cccc(F)c1)CN1C(=O)c2ccccc2C1=O', 'Prediction2 : NC(Cc1ccc(F)cc1)CN1C(=O)c2ccccc2C1=O', {'maccs': 1.0, 'rdkit4': 0.9838709677419355, 'aes': 0.9545454545454546}],
- ['Product : CCOC(=O)CCCC(=O)c1ccc(F)c2ccccc12', 'Reactant1 : CCOC(=O)CCCC(=O)O', 'Reactant2 : Fc1cccc2ccccc12', 'Prediction2 :

```

Fc1cccc2c(F)cccc12', {'maccs': 1.0, 'rdkit4': 1.0, 'aes':
0.8888888888888888}],

['Product : CCOC(=O)CCCC(=O)c1ccc(F)c2ccccc12', 'Reactant1 : CCOC(=O)
)CCCC(=O)O', 'Reactant2 : Fc1cccc2ccccc12', 'Prediction2 :
Fc1ccc(F)c2ccccc12', {'maccs': 1.0, 'rdkit4': 1.0, 'aes':
0.8888888888888888}],

['Product : COc1ccccc1N1CCN(C(=O)c2ccc(-c3cnc4c(c3)N(Cc3cc(Cl)ccc3C(F)
(F)F)CCN4)cc2)CC1', 'Reactant1 : COc1ccccc1N1CCNCC1', 'Reactant2 :
O=C(O)c1ccc(-c2cnc3c(c2)N(Cc2cc(Cl)ccc2C(F)(F)F)CCN3)cc1', '
Prediction2 : O=C(O)c1ccc(-c2cnc3c(c2)N(Cc2cc(Cl)ccc2C(F)(F)F)
CCN3)c1', {'maccs': 1.0, 'rdkit4': 0.9905660377358491, 'aes':
0.9722222222222222}]
]

```

Ten double mutant (DM) examples.

```

['Product : CCCCc1ccc(CN(C(=O)C=Cc2cccc(C(F)(F)F)c2)C(Cc2ccccc2)C(=O)
N2CCN(Cc3ccccc3)CC2)cc1', 'Reactant1 : CCCCc1ccc(CNC(Cc2ccccc2)C
(=O)N2CCN(Cc3ccccc3)CC2)cc1', 'Reactant2 : O=C(O)C=Cc1ccc(C(F)(F)
F)c1', 'Prediction2 : O=C(O)C=Cc1ccc(C(F)(F)F)cc1', {'maccs': 1.0,
'rdkit4': 0.9705882352941176, 'aes': 0.9}],

['Product : O=C(Nc1cccc(-c2nn3ccccc3c2-c2ccnc(Nc3cccc(OCCN4CCOCC4)c3)
n2)c1)c1c(F)cccc1F', 'Reactant1 : O=C(Nc1cccc(-c2nn3ccccc3c2-
c2ccnc(Cl)n2)c1)c1c(F)cccc1F', 'Reactant2 : Nc1cccc(OCCN2CCOCC2)c1
', 'Prediction2 : Nc1ccc(OCCN2CCOCC2)cc1', {'maccs':
0.9767441860465116, 'rdkit4': 0.9705882352941176, 'aes':
0.9047619047619048}],

['Product : CC(C)C(=O)Nc1cccc(C2CCN(CCCCC(=O)c3ccc(Cl)cc3)CC2)c1', '
Reactant1 : CC(C)C(=O)Nc1cccc(C2CCNCC2)c1', 'Reactant2 : O=C(
CCCCCl)c1ccc(Cl)cc1', 'Prediction2 : O=C(CCCCl)c1cccc(Cl)c1', {'
maccs': 0.9375, 'rdkit4': 0.9459459459459459, 'aes':
0.8888888888888888}],

['Product : CC(=O)SCC(Cc1ccccc1)C(=O)Nc1cccc(C(=O)O)c1', 'Reactant1 :
CC(=O)SCC(Cc1ccccc1)C(=O)Cl', 'Reactant2 : Nc1cccc(C(=O)O)c1', '
Prediction2 : Nc1ccc(C(=O)O)cc1', {'maccs': 1.0, 'rdkit4': 0.96, '
aes': 0.8666666666666667}],

['Product : C#CCOc1ccc(C(=O)O)c(OCC#C)c1', 'Reactant1 : C#
CCOc1ccccc1C(=O)O', 'Reactant2 : C#CCOc1ccc(C(=O)O)c1', '
Prediction2 : C#CCOc1ccc(C(=O)O)cc1', {'maccs': 1.0, 'rdkit4':

```

0.9714285714285714, 'aes': 0.9047619047619048]],

['Product : O=C(O)C(CC1CCCC1)N1CC(CN2CCC(CCCc3ccc(F)cc3)CC2)C(c2cccc(F)c2)C1', 'Reactant1 : O=CC1CN(C(CC2CCCC2)C(=O)O)CC1c1cccc(F)c1', 'Reactant2 : Fc1ccc(CCCC2CCNCC2)cc1', 'Prediction2 : Fc1cc(F)cc(CCCC2CCNCC2)c1', {'maccs': 1.0, 'rdkit4': 0.9411764705882353, 'aes': 0.8947368421052632}],

['Product : CC(C)C(=O)Nc1cccc(C2CCN(CCCCC(=O)c3ccc(I)cc3)CC2)c1', 'Reactant1 : CC(C)C(=O)Nc1cccc(C2CCNCC2)c1', 'Reactant2 : O=C(CCCCCI)c1ccc(I)cc1', 'Prediction2 : O=C(CCCCCI)c1cccc(I)c1', {'maccs': 1.0, 'rdkit4': 0.972972972972973, 'aes': 0.8947368421052632}],

['Product : COc1cc(OC)nc(Oc2ccccc2C=NOCC=Cc2cccc(Cl)c2)n1', 'Reactant1 : COc1cc(OC)nc(Oc2ccccc2C=NO)n1', 'Reactant2 : Clc1cccc(C=CCBr)c1', 'Prediction2 : Clc1ccc(C=CCBr)cc1', {'maccs': 1.0, 'rdkit4': 0.9583333333333334, 'aes': 0.875}],

['Product : O=C(CNc1cccc(F)c1)Nc1cccc1C(=O)O', 'Reactant1 : O=C(CBr)Nc1cccc1C(=O)O', 'Reactant2 : Nc1cccc(F)c1', 'Prediction2 : Nc1ccc(F)cc1', {'maccs': 1.0, 'rdkit4': 0.9285714285714286, 'aes': 0.8181818181818182}],

['Product : CCOC(=O)c1ccc2c(C(=O)NCc3cc(F)cc(F)c3)c(C(C)C)n(Cc3ccccc3)c2c1', 'Reactant1 : CCOC(=O)c1ccc2c(C(=O)O)c(C(C)C)n(Cc3ccccc3)c2c1', 'Reactant2 : NCc1cc(F)cc(F)c1', 'Prediction2 : NCc1ccc(F)cc1', {'maccs': 1.0, 'rdkit4': 0.8947368421052632, 'aes': 0.8333333333333334}]

]

References

- [1] Robert D. Brown and Yvonne C. Martin. "The information content of 2D and 3D structural descriptors relevant to ligand-receptor binding". In: *Journal of Chemical Information and Computer Sciences* 37.1 (1997), pp. 1–9. ISSN: 00952338. DOI: 10.1021/ci960373c.
- [2] David E. Patterson et al. "Neighborhood behavior: A useful concept for validation of 'molecular diversity' descriptors". In: *Journal of Medicinal Chemistry* 39.16 (1996), pp. 3049–3059. ISSN: 00222623. DOI: 10.1021/jm960290n.
- [3] John S. Delaney. "Assessing the ability of chemical similarity measures to discriminate between active and inactive compounds". In: *Molecular Diversity* 1.4 (1996), pp. 217–222. ISSN: 13811991. DOI: 10.1007/BF01715525.

- [4] Hans Matter. "Selecting optimally diverse compounds from structure databases: A validation study of two-dimensional and three-dimensional molecular descriptors". In: *Journal of Medicinal Chemistry* 40.8 (1997), pp. 1219–1229. ISSN: 00222623. DOI: 10.1021/jm960352+.
- [5] R. D. Brown and Y. C. Martin. "An evaluation of structural descriptors and clustering methods for use in diversity selection." In: *SAR and QSAR in environmental research* 8.1-2 (1998), pp. 23–39. ISSN: 1062936X. DOI: 10.1080/10629369808033260.
- [6] Yvonne C. Martin, James L. Kofron, and Linda M. Traphagen. "Do structurally similar molecules have similar biological activity?" In: *Journal of Medicinal Chemistry* 45.19 (2002), pp. 4350–4358. ISSN: 00222623. DOI: 10.1021/jm020155c.
- [7] Steven W. Muchmore et al. "Application of belief theory to similarity data fusion for use in analog searching and lead hopping". In: *Journal of Chemical Information and Modeling* 48.5 (2008), pp. 941–948. ISSN: 15499596. DOI: 10.1021/ci7004498.
- [8] Mathias Dunkel et al. "SuperPred: drug classification and target prediction." In: *Nucleic acids research* 36.Web Server issue (2008), pp. 55–59. ISSN: 13624962. DOI: 10.1093/nar/gkn307.
- [9] Jürgen Bajorath et al. "Activity-relevant similarity values for fingerprints and implications for similarity searching". In: *F1000Research* 5.0 (2016). ISSN: 1759796X. DOI: 10.12688/f1000research.8357.1.
- [10] Martin Thimm et al. "Comparison of 2D similarity and 3D superposition. Application to searching a conformational drug database". In: *Journal of Chemical Information and Computer Sciences* 44.5 (2004), pp. 1816–1822. ISSN: 00952338. DOI: 10.1021/ci049920h.
- [11] Martin Vogt and Jürgen Bajorath. "Introduction of a generally applicable method to estimate retrieval of active molecules for similarity searching using fingerprints". In: *ChemMedChem* 2.9 (2007), pp. 1311–1320. ISSN: 18607179. DOI: 10.1002/cmdc.200700090.
- [12] Anne Mai Wassermann, Eugen Lounkine, and Meir Glick. "Bioturbo similarity searching: Combining chemical and biological similarity to discover structurally diverse bioactive molecules". In: *Journal of Chemical Information and Modeling* 53.3 (2013), pp. 692–703. ISSN: 15499596. DOI: 10.1021/ci300607r.
- [13] Igor V. Tetko et al. "State-of-the-art augmented NLP transformer models for direct and single-step retrosynthesis". In: *Nature Communications* 11.1 (2020), pp. 1–24. ISSN: 20411723. DOI: 10.1038/s41467-020-19266-y. arXiv: 2003.02804. URL: <http://dx.doi.org/10.1038/s41467-020-19266-y>.
- [14] Andrew Jaegle et al. "Perceiver: General Perception with Iterative Attention". In: (2021). arXiv: 2103.03206. URL: <http://arxiv.org/abs/2103.03206>.
- [15] Pavel Karpov, Guillaume Godin, and Igor V. Tetko. "A Transformer Model for Retrosynthesis". In: *Artificial Neural Networks and Machine Learning – ICANN 2019: Workshop and Special Sessions*. Cham: Springer International Publishing, 2019, pp. 817–830. ISBN: 978-3-030-30493-5.

- [16] Kangjie Lin et al. “Automatic retrosynthetic route planning using template-free models”. In: *Chemical Science* 11.12 (2020), pp. 3355–3364. ISSN: 20416539. DOI: 10.1039/c9sc03666k.
- [17] Philippe Schwaller et al. “Predicting retrosynthetic pathways using transformer-based models and a hyper-graph exploration strategy”. In: *Chemical Science* 11.12 (2020), pp. 3316–3325. ISSN: 20416539. DOI: 10.1039/c9sc05704h.
- [18] Marwin H.S. Segler, Mike Preuss, and Mark P. Waller. “Planning chemical syntheses with deep neural networks and symbolic AI”. In: *Nature* 555.7698 (2018), pp. 604–610. ISSN: 14764687. DOI: 10.1038/nature25978.
- [19] Samuel Genheden et al. “AiZynthFinder: a fast, robust and flexible open-source software for retrosynthetic planning”. In: *Journal of Cheminformatics* 12.1 (2020), pp. 1–9. ISSN: 17582946. DOI: 10.1186/s13321-020-00472-1.
- [20] Marwin H.S. Segler and Mark P. Waller. “Neural-Symbolic Machine Learning for Retrosynthesis and Reaction Prediction”. In: *Chemistry - A European Journal* 23.25 (2017), pp. 5966–5971. ISSN: 15213765. DOI: 10.1002/chem.201605499.
- [21] Wengong Jin et al. “Predicting organic reaction outcomes with weisfeiler-lehman network”. In: *Adv. Neur. In. 2017-Decem.Nips* (2017), pp. 2608–2617. ISSN: 10495258.
- [22] Sara Szymkuć et al. *Computer-Assisted Synthetic Planning: The End of the Beginning*. Vol. 55. 20. 2016, pp. 5904–5937. ISBN: 1671471555. DOI: 10.1002/anie.201506101.
- [23] Connor W. Coley et al. “Prediction of Organic Reaction Outcomes Using Machine Learning”. In: *ACS Central Science* 3.5 (2017), pp. 434–443. ISSN: 23747951. DOI: 10.1021/acscentsci.7b00064.
- [24] James Law et al. “Route Designer: A Retrosynthetic Analysis Tool Utilizing Automated Retrosynthetic Rule Generation”. In: *J. Chem. Inf. Model.* 49.3 (2009), pp. 593–602. DOI: 10.1021/ci800228y.
- [25] Bowen Liu et al. “Retrosynthetic Reaction Prediction Using Neural Sequence-to-Sequence Models”. In: *ACS Central Science* 3.10 (2017), pp. 1103–1113. ISSN: 23747951. DOI: 10.1021/acscentsci.7b00303.
- [26] Miguel Domingo et al. “How Much Does Tokenization Affect Neural Machine Translation?” In: (2018). ISSN: 2331-8422. arXiv: 1812.08621. URL: <http://arxiv.org/abs/1812.08621>.
- [27] Dávid Péter Kovács, William McCorkindale, and Alpha A. Lee. “Quantitative interpretation explains machine learning models for chemical reaction prediction and uncovers bias”. In: *Nature Communications* 12.1 (2021), pp. 1–9. ISSN: 20411723. DOI: 10.1038/s41467-021-21895-w. URL: <http://dx.doi.org/10.1038/s41467-021-21895-w>.
- [28] Shuangjia Zheng et al. “Predicting Retrosynthetic Reactions Using Self-Corrected Transformer Neural Networks”. In: *Journal of Chemical Information and Modeling* 60.1 (2020), pp. 47–55. DOI: 10.1021/acs.jcim.9b00949. URL: <https://doi.org/10.1021/acs.jcim.9b00949>.

- [29] Hongliang Duan et al. "Retrosynthesis with attention-based NMT model and chemical analysis of "wrong" predictions". In: *RSC Advances* 10.3 (2020), pp. 1371–1378. ISSN: 20462069. DOI: 10.1039/c9ra08535a.
- [30] Eunji Kim et al. "Valid, Plausible, and Diverse Retrosynthesis Using Tied Two-Way Transformers with Latent Variables". In: *Journal of Chemical Information and Modeling* 61.1 (2021), pp. 123–133. ISSN: 15205142. DOI: 10.1021/acs.jcim.0c01074.
- [31] Noel M. O'Boyle and Andrew Dalke. "DeepSMILES: An adaptation of SMILES for use in machine-learning of chemical structures". In: *ChemRxiv* (2018), pp. 1–9. ISSN: 2573-2293. DOI: 10.26434/chemrxiv.7097960.
- [32] Mario Krenn et al. "Self-referencing embedded strings (SELFIES): A 100% robust molecular string representation". In: *Machine Learning: Science and Technology* 1.4 (2020), p. 045024. ISSN: 2632-2153. DOI: 10.1088/2632-2153/aba947.
- [33] D. M. Lowe. "Extraction of chemical structures and reactions from the literature". In: (2012). DOI: 10.17863/CAM.16293.
- [34] Daniel Lowe. "Chemical reactions from US patents (1976-Sep2016)". In: (2017). DOI: 10.6084/m9.figshare.5104873.v1. URL: https://figshare.com/articles/Chemical%7B%5C_%7Dreactions%7B%5C_%7Dfrom%7B%5C_%7DUS%7B%5C_%7Dpatents%7B%5C_%7D1976-Sep2016%7B%5C_%7D/5104873.
- [35] Philippe Schwaller et al. "'Found in Translation': predicting outcomes of complex organic chemistry reactions using neural sequence-to-sequence models". In: *Chem. Sci.* 9.28 (2018), pp. 6091–6098. ISSN: 20416539. DOI: 10.1039/c8sc02339e.
- [36] Taku Kudo and John Richardson. "SentencePiece: A simple and language independent subword tokenizer and detokenizer for neural text processing". In: *EMNLP 2018 - Conference on Empirical Methods in Natural Language Processing: System Demonstrations, Proceedings* (2018), pp. 66–71. DOI: 10.18653/v1/d18-2012. arXiv: 1808.06226.
- [37] Greg Landrum. *RDKit: Open-Source Cheminformatics Software*. 2016. URL: https://github.com/rdkit/rdkit/releases/tag/Release_2020_03_1.
- [38] Adam Paszke et al. "PyTorch: An Imperative Style, High-Performance Deep Learning Library". In: *Advances in Neural Information Processing Systems* 32. Ed. by H. Wallach et al. Curran Associates, Inc., 2019, pp. 8024–8035.
- [39] Amol Thakkar et al. "Datasets and their influence on the development of computer assisted synthesis planning tools in the pharmaceutical domain". In: *Chemical Science* 11.1 (2020), pp. 154–168. ISSN: 20416539. DOI: 10.1039/c9sc04944d.
- [40] Ralf Fick, Wolf-Dietrich Ihlenfeldt, and Johann Gasteiger. "Computer-Assisted Design of Syntheses for Heterocyclic Compounds". In: *Heterocycles* 40.2 (1995), pp. 993–1007.
- [41] Barbara Mikulak-Klucznik et al. "Computational planning of the synthesis of complex natural products". In: *Nature* 588.7836 (2020), pp. 83–88. ISSN: 14764687. DOI: 10.1038/s41586-020-2855-y.

- [42] Jennifer N Wei, David Duvenaud, and Alán Aspuru-Guzik. “Neural Networks for the Prediction of Organic Chemistry Reactions”. In: *ACS Cent. Sci.* 2.10 (2016), pp. 725–732. DOI: 10.1021/acscentsci.6b00219.
- [43] Marwin H.S. Segler and Mark P. Waller. “Modelling Chemical Reasoning to Predict and Invent Reactions”. In: *Chem. Eur. J.* 23.25 (2017), pp. 6118–6128. ISSN: 15213765. DOI: 10.1002/chem.201604556.

REVIEWERS' COMMENTS

Reviewer #1 (Remarks to the Author):

The authors have done a detailed job at answering the reviewer's comments, however the final version still fails to convince on two points:

1) compared to the recent paper of the same authors, which they cite as ref. no. 39, <https://jcheminf.biomedcentral.com/track/pdf/10.1186/s13321-020-00482-z.pdf>

the present manuscript still really lacks convincing novelty, the idea of using fragment information for transformer models was already there, the only thing that changed is the way the fragments are encoded. This is an interesting variation, but conceptually this does not make a huge change

2) the top-1 accuracy is really not relevant in the perspective of multi-step retrosynthesis, and using this metrics, where their model is slightly better over others, is simply not a good choice because any retrosynthesis approach must make sense towards a mutli-step application

For the above reasons, the present paper lacks general interest and is not suitable for a broad readership as that of Nature Communications. The work should be published in a specialized journal.

Reviewer #2 (Remarks to the Author):

Dear authors,

thanks for answering all my questions. I believe the article is more consistent now. I will recommend the paper for publishing.

Reviewer #3 (Remarks to the Author):

The authors have made significant improvements to the manuscript

I'm still sceptical about the value of the bioactivity similarity concept, however, I don't have any objections that it is published and assessed by the scientific community